# The ribosome derives the energy to translocate and unwind mRNA from EF-G binding

Hossein Amiri [1,2,3,4,11] ✉, William J. Van Patten[1,2,5,11], Gillian Rexroad[6,7], Varsha P. Desai[1,2], Benjamen A. Sterwerf[1,2], Laura Lancaster[6,7], Harry F. Noller[6,7] ✉ & Carlos Bustamante [1,2,4,5,8,9,10] ✉

The GTPase EF-G catalyzes translocation of mRNA and tRNAs relative to the ribosome and helps maintain the reading frame during protein synthesis. Which events directly require EF-G-mediated GTP hydrolysis during translocation are still debated. Using high-resolution optical tweezers endowed with single-molecule fluorescence detection, we simultaneously monitored binding of fluorescently-labeled EF-G to ribosomes and either mRNA unwinding or mRNA translocation relative to the body domain of the small ribosomal subunit. Using EF-G mutants and GTP analogs, we find that neither mRNA unwinding nor translocation require GTP hydrolysis and that these are independent events that may or may not temporally coincide. We propose that "tight binding" of EF-G to the ribosome triggers mRNA unwinding and translocation of mRNA relative to the 30S body domain and that while GTP hydrolysis kinetically accelerates translocation, it is thermodynamically required only to liberate the tightly bound EF-G from the ribosome.

During the elongation phase of protein synthesis, the ribosome translocates mRNA in single-codon (3 nt) steps[1,2]. This coordinated translocation of the mRNA and tRNAs is catalyzed by the GTPase elongation factor EF-G and guided by conformational changes of the large and small ribosomal subunits following peptide bond formation[3–10]. Specifically, the 30S subunit rotates with respect to the 50S subunit, which moves the tRNAs on the 50S subunit resulting in hybrid tRNA states[11]. Next, forward rotation or "swivel" of the 30S head domain, accompanied by a partial reversion of inter-subunit rotation, moves the mRNA and tRNAs relative to the body domain to produce chimeric hybrid tRNA states[12,13]. This step is followed by the reverse 30S head rotation without carrying back the mRNA and tRNAs, which together with full reverse inter-subunit rotation, completes the mRNA/

tRNA movement and resets the ribosome to the classical post-translocation state.

Translocation involves directional mRNA movement in precise codon steps and unwinding of mRNA secondary structures[2,7,14]. In the absence of an external energy source, these processes are endergonic ($\Delta G > 0$) and will not occur spontaneously. To drive these processes, the ribosome must couple them with exergonic reactions, which could include peptidyl transfer between P- and A-site tRNAs and EF-G-mediated GTP hydrolysis. After GTP hydrolysis, EF-G retains the cleaved γ-phosphate for some time[15,16] before releasing it as inorganic phosphate ($P_i$), which contributes to the free energy available from hydrolysis. Two roles for GTP hydrolysis and $P_i$ release by EF-G during translocation have been described. In the first role, the energy from

[1]Institute for Quantitative Biosciences-QB3, University of California, Berkeley, CA, USA. [2]Jason L. Choy Laboratory of Single-Molecule Biophysics, University of California, Berkeley, CA, USA. [3]Department of Molecular and Cell Biology, University of California, Berkeley, CA, USA. [4]Howard Hughes Medical Institute, University of California, Berkeley, CA, USA. [5]Biophysics Graduate Group, University of California, Berkeley, CA, USA. [6]Center for Molecular Biology of RNA, University of California, Santa Cruz, CA, USA. [7]Department of Molecular, Cell and Developmental Biology, University of California, Santa Cruz, CA, USA. [8]Department of Chemistry, University of California, Berkeley, CA, USA. [9]Department of Physics, University of California, Berkeley, CA, USA. [10]Kavli Energy Nanoscience Institute, University of California, Berkeley, CA, USA. [11]These authors contributed equally: Hossein Amiri, William J. Van Patten.
✉e-mail: mamiri@berkeley.edu; harry@nuvolari.ucsc.edu; carlosb@berkeley.edu

GTP hydrolysis and $P_i$ release is suggested to be directly converted to mechanical work[4,15,17,18]. In the second role, this energy is instead used for reducing the affinity of EF-G to the ribosome, allowing for EF-G release and the resetting of the translational cycle[9,19–25]. Recently, three independent cryo-electron microscopy (cryo-EM) studies in the presence[18,23] and absence[22] of inhibitors investigated the structures of ribosomal translocation intermediates. These structures show distinct modes of binding of EF-G to the ribosome over the course of translocation, differing in the conformational state of the ribosome and in the presence or absence of the γ-phosphate of GTP. Specifically, $P_i$ release is seen concomitant with conformational changes that move the mRNA relative to the 30S body domain. Strictly speaking, it is not possible to establish a causal relationship between these two correlated events from structural snapshots. Indeed, the three studies did not agree on when GTP hydrolysis and $P_i$ release are required during translocation.

Ensemble kinetics experiments have indicated that GTP hydrolysis occurs early in the process of translocation[4,26]. Importantly, ribosomes incubated with EF-G and non-hydrolyzable GTP analogs are able to perform single-turnover translocation, albeit at slower rates and with lower efficiency than with GTP[4,16,21,24,27,28]. Likewise, the antibiotic sparsomycin induces single-turnover mRNA unwinding and 30S translocation in the absence of EF-G altogether[14,29]. Moreover, multi-turnover translation can also occur in the absence of EF-G, albeit at very slow rates and under special conditions (e.g. addition of thiol-modifying reagents)[30,31], indicating that the peptidyl transfer energy suffices to drive translocation, at least on unstructured mRNA. GTP hydrolysis and $P_i$ release by EF-G further contribute to the overall free energy balance and may be essential for multi-turnover translation and unwinding of mRNAs that contain stable secondary structures. However, the fundamental question of whether GTP hydrolysis and $P_i$ release are required in advance to drive the energetically uphill mechanical events (e.g., mRNA unwinding) during each cycle of ribosomal translocation is still a matter of debate[17,24].

Here, we investigate the requirement for EF-G-mediated GTP hydrolysis and $P_i$ release during translocation by using high-resolution dual-trap optical tweezers endowed with single-molecule fluorescence detection capabilities ("fleezers"). This instrument enables simultaneous monitoring of binding of fluorescent EF-G to the ribosome and ribosomal unwinding of a hairpin. By perturbing the system with EF-G mutants, a ribosomal protein mutant, or GTP analogs, we show that neither GTP hydrolysis nor $P_i$ release is required for the endergonic hairpin unwinding step of translocation. Furthermore, by simultaneously monitoring hairpin unwinding and changes in Förster resonance energy transfer (FRET) between the 30S head and body domains, we find that forward 30S head rotation (a process that involves movement of the mRNA and tRNAs relative to the 30S body domain) can temporally coincide with unwinding. However, using an alternative fleezers assay that simultaneously monitors fluorescently-labeled EF-G binding to the ribosome and direct mRNA translocation relative to the 30S body, we find that translocation can, under GTPase perturbations, be uncoupled from hairpin unwinding and occur after it. Importantly, this alternative assay also shows that mRNA translocation relative to the 30S body does not require GTP hydrolysis, although it is kinetically accelerated by it. We conclude that EF-G-mediated GTP hydrolysis is not required in ribosomal translocation until after mRNA translocation relative to the 30S body has occurred. We propose that the energy derived from binding of EF-G to the ribosome and intrinsic ribosome conformational changes drive the early endergonic events of the translation cycle, including hairpin unwinding and mRNA translocation relative to the 30S body. We argue that $P_i$ release provides instead the drop in free energy required to dissociate EF-G and to initiate reverse 30S head rotation, thus completing the translocation cycle.

## Results

### Simultaneous monitoring of hairpin unwinding and EF-G binding

To investigate the requirement for EF-G-mediated GTP hydrolysis in mRNA unwinding, we used an optical tweezers assay in which an mRNA hairpin is tethered between two optically trapped polystyrene beads using DNA handles (Fig. 1a). During each translocation cycle, the ribosome unwinds the hairpin by one codon (3 bp) measured by the optical traps, while the binding of Cy3-labeled EF-G to the ribosome is co-temporally monitored by the confocal single-molecule fluorescence detection capability of the fleezers instrument[32] (Fig. 1b, Supplementary Fig. 1). Using this assay, we measured the time elapsed between EF-G binding and hairpin unwinding ($\tau_{unwinding}$) and the time between unwinding and EF-G release ($\tau_{release}$) (Fig. 1c). Under moderately high tether pulling forces (~14pN) and in the presence of 1 mM GTP, the observed distributions for these parameters yielded an overall characteristic time estimate of $72 \pm 45$ ms for $\tau_{unwinding}$ and $380 \pm 230$ ms for $\tau_{release}$, similar to those found previously[32] (Fig. 1c, top green bar, Fig. 1d, e, magenta dots, see Methods). The distributions appear bi-exponential, indicating that unwinding and release follow bifurcated paths, the causes of which are unknown.

It is not known how perturbing EF-G-mediated GTP hydrolysis and $P_i$ release affect $\tau_{unwinding}$ and $\tau_{release}$. The answer will reveal when these chemical events are required in the translocation cycle. For example, if GTP hydrolysis is slowed and $\tau_{unwinding}$ is lengthened as a result, then hydrolysis is likely required for hairpin unwinding and occurs before or concurrently with it. Alternatively, if only $\tau_{release}$ is lengthened, then hydrolysis must not be required for hairpin unwinding but must be required instead for EF-G unbinding. (Supplementary Fig. 2).

### Hairpin unwinding by the ribosome does not require EF-G-mediated GTP hydrolysis or $P_i$ release

We first tested whether GTP hydrolysis is required for hairpin unwinding by using a translation solution containing fluorescently labeled WT EF-G and a mixture of GTP and a GTP analog. Using only the analog and no GTP is unfeasible for this multi-turnover assay, because the other essential elongation factor, EF-Tu, also requires GTP.

Total EF-G dwell times ($\tau_{unwinding} + \tau_{release}$) for a mixture of 1 mM GTP and 0.5 mM of the non-hydrolyzable GDPNP varied from hundreds of milliseconds to tens of seconds (Supplementary Fig. 3a), in contrast to WT EF-G dwell times which are typically under 500 ms. Importantly, the distribution of $\tau_{unwinding}$ in the presence of the mixture of GTP and GDPNP shows no significant change compared to that of GTP alone, whereas $\tau_{release}$ is lengthened considerably (Fig. 1d, e, blue dots). Compared to the GTP-only condition (Fig. 1e, magenta dots, dashed line), the distribution of $\tau_{release}$ for the GTP + GDPNP mixture contains a slow population likely associated with GDPNP events (Fig. 1c, third row, Fig. 1e, blue dots, dashed line). This population has a characteristic $\tau_{release}$ fit value of $8.5 \pm 1.5$ s (Fig. 1c), over an order of magnitude longer than that of the fast (GTP) population.

Similarly, using a mixture of 1 mM GTP and 0.5 mM of slowly-hydrolyzable GTPγS increased the total EF-G dwell time, although to a lesser extent than with GDPNP (Supplementary Fig. 3b). Again, $\tau_{unwinding}$ for the GTP + GTPγS mixture remains unchanged compared to the GTP-only condition, whereas the distribution of $\tau_{release}$ contains a slow population with a characteristic time of $1.8 \pm 0.2$ s, almost an order of magnitude longer than that of GTP (Fig. 1c, bottom row, Fig. 1d, e, purple dots, dashed line).

GDPNP is essentially non-hydrolyzable at our experimental timescale and practically limits translocation to a single turnover[4,24]. However, when using a GTP + GDPNP mixture, we observed processive multi-turnover translocation with interspersed normal and lengthened EF-G binding events that are productive (Supplementary Fig. 3a). We rule out the possibility of exchange of EF-G·GDPNP for EF-G·GTP on the

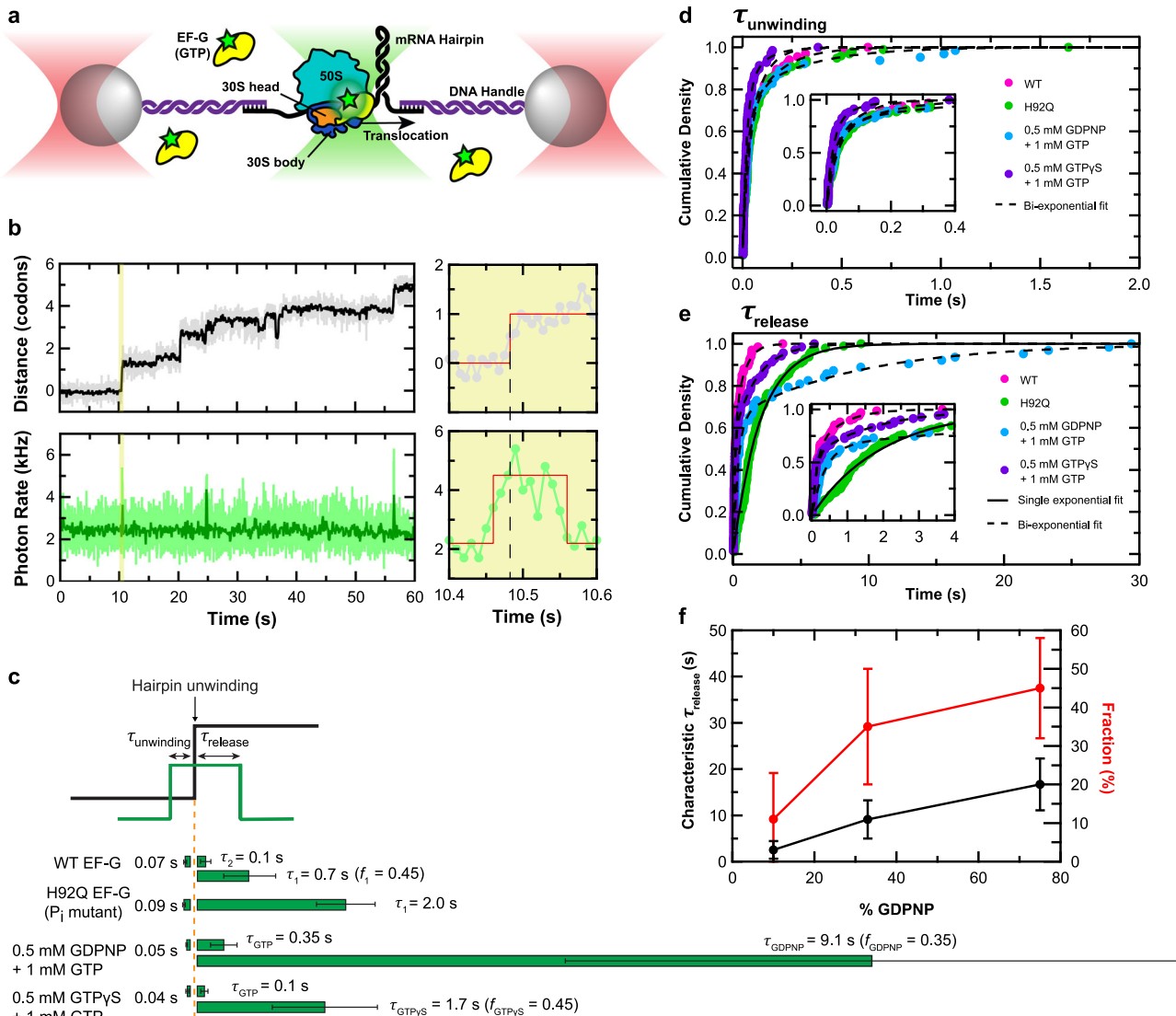

**Fig. 1 | Dependence of hairpin unwinding on GTP hydrolysis and $P_i$ release.**
**a** Schematic of the "fleezers" hairpin unwinding assay setup for simultaneous detection of mRNA hairpin unwinding and EF-G binding during ribosomal translocation. **b** A fleezers trajectory of consecutive steps taken by a single ribosome in the presence of wild-type EF-G and GTP, showing stepwise opening of the hairpin in the optical tweezers (top) channel and corresponding EF-G binding events with elevated photon emission rates in the fluorescence (bottom) channel. The pale and dark lines correspond to raw and 10-point smoothed data, respectively. The shaded area is magnified in the right panels, in which red lines demarcate the detected transition times. **c** Summary of $\tau_{unwinding}$ and $\tau_{release}$ measurements for WT EF-G, H92Q EF-G, the GDPNP condition, and the GTPγS condition. For $\tau_{unwinding}$, mean and standard error are shown. For $\tau_{release}$ characteristic lifetimes and 95% confidence interval from fits are shown. For analogs, the top and bottom green bars for

$\tau_{release}$ correspond to rate estimates for the fast (GTP) and slow (analog) populations, respectively, at the indicated fraction ($f$) estimates. All sample sizes are listed in Supplemental Tables 1,2. **d, e** Cumulative distribution plots for $\tau_{unwinding}$ (**d**) and $\tau_{release}$ (**e**) in the hairpin unwinding assay for conditions listed in panel C. Single- and bi-exponential fits are shown as solid and dashed curves, respectively. The inset in each plot shows a magnified view for the shorter observed times. While the distribution of $\tau_{unwinding}$ remains unchanged under the different conditions tested, $\tau_{release}$ is lengthened in the presence of the $P_i$ release mutant of EF-G or with GTP analogs. **f** Characteristic $\tau_{release}$ time (black line, left axis) and fraction (red line, right axis) of the slow population from the bi-exponential fits of $\tau_{release}$ across the range of GDPNP percentage tested. Error bars represent 95% confidence interval. Sample sizes are listed in Supplemental Table 2. Source data are provided as a Source Data file.

ribosome since we do not detect unbinding and re-binding of EF-G during the long dwell times. This observation leaves only two possibilities to explain the multi-turnover unwinding displaying lengthened but finite EF-G dwell times: either GDPNP is eventually exchanged for GTP on the ribosome-bound EF-G, allowing translocation to resume after only a delay, or EF-G·GDPNP is eventually released from the ribosome after productive stepping without any hydrolysis. As mentioned above, the $\tau_{release}$ distribution for the GTP + GDPNP condition consists of a fast (GTP) and a slow (GDPNP) population. If nucleotide exchange occurs, we expect that altering the GDPNP:GTP ratio would change not only the slow population fraction in the $\tau_{release}$ distribution, but also the characteristic release time for this population (which

would be limited by a ratio-dependent rate of exchange of GDPNP and GTP). By contrast, if the long release times are due entirely to EF-G·GDPNP (without exchange), altering the ratio should only change the slow fraction but not its characteristic time. To distinguish between these possibilities, we performed a titration of GDPNP:GTP ratio while keeping the total nucleotide concentration constant. Increasing this ratio from 1:9 to 3:1 increased both the characteristic time (from $2.5 \pm 1.9$ s to $17 \pm 5.6$ s) and the fraction estimate (from $0.11 \pm 0.1$ to $0.45 \pm 0.1$) of the slow population for $\tau_{release}$ (Fig. 1f, Supplementary Fig. 3c). This result supports the hypothesis that EF-G can exchange its bound nucleotide while remaining bound to the ribosome. While nucleotide exchange by ribosome-bound EF-G was proposed

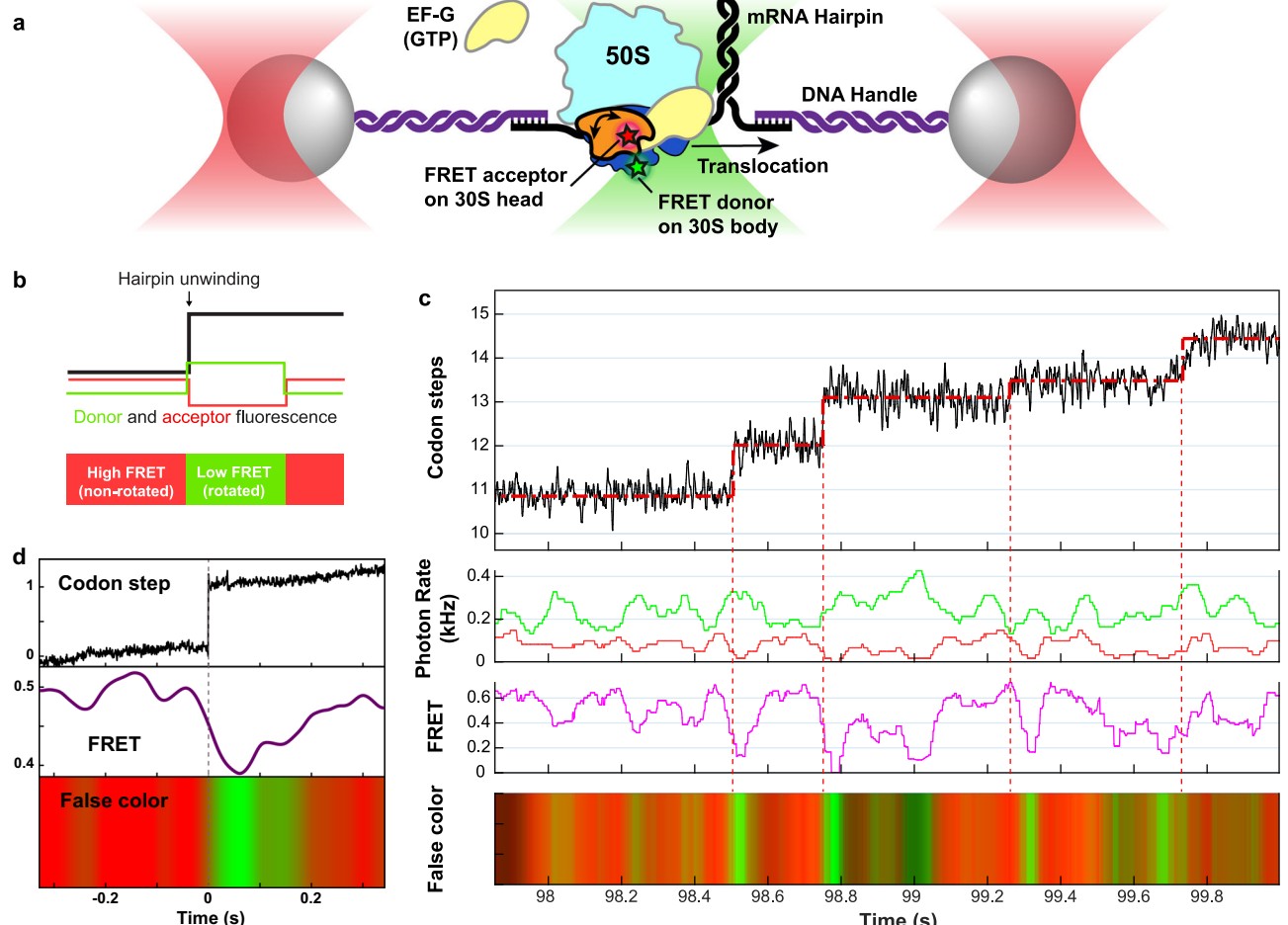

**Fig. 2 | Temporal correlation of hairpin unwinding and 30S head rotation.**
**a** Schematic of the fleezers hairpin unwinding setup with FRET for simultaneous detection of mRNA hairpin unwinding and ribosomal 30S head rotation during translocation. **b** Expected pattern of donor and acceptor fluorescence if hairpin unwinding coincides with forward head rotation. **c** A fleezers trajectory showing consecutive unwinding steps (first panel), anticorrelated fluorescence in the green and red channels (second panel), calculated ratiometric FRET efficiency (third panel), and FRET visualized by color (last panel, see "Methods"). **d** Filtered event average from many steps ($n = 396$) showing red-to-green (high-to-low FRET) transition around the time of hairpin unwinding. Source data are provided as a Source Data file.

previously[27], our data suggest that such exchange is very slow (seconds timescale). Whether EF-Tu·GTP·aa-tRNA can also exchange its GTP for GDPNP under these conditions remains to be examined.

To complement the experiments performed with GTP analogs, we also perturbed GTP hydrolysis by using the H92A mutant of EF-G which is highly deficient in both GTP hydrolysis and $P_i$ release[33]. In the presence of GTP, $\tau_{unwinding}$ for this mutant remained unchanged relative to WT while $\tau_{release}$ increased dramatically (~20 s), occasionally exceeding 50 s (Supplementary Fig. 4b, c). Nonetheless, we observe multi-turnover translocation with this mutant.

If $\tau_{unwinding}$ is insensitive to perturbed GTP hydrolysis, it is expected to also be insensitive to perturbed $P_i$ release. We therefore tested the H92Q mutant of EF-G which has been reported to exhibit a 25-fold reduced rate of $P_i$ release compared to wild type (WT) without significantly affecting the rate of GTP hydrolysis[33]. We observed greatly lengthened H92Q EF-G dwell times that extended to multiple seconds (Supplementary Fig. 5). As expected, fitting the distributions of $\tau_{unwinding}$ and $\tau_{release}$ for H92Q EF-G clearly revealed an unchanged $\tau_{unwinding}$ compared to WT EF-G, but a significantly lengthened $\tau_{release}$ (2.0 ± 0.4 s) (Fig. 1c, second row, Fig. 1d, e, green dots). Furthermore, the distribution of $\tau_{release}$ for the H92Q mutant, unlike that of the WT, is well described by a single-exponential fit (Fig. 1e, solid line), suggesting that $P_i$ release in these conditions has become the overall rate-limiting step of EF-G unbinding from the ribosome.

To further verify that $P_i$ release is not required for hairpin unwinding, we tested reconstituted ribosomes containing the V67D mutant of ribosomal protein L7/L12, which was shown to slow down $P_i$ release from EF-G[34]. As in the case of H92Q EF-G, we found that using WT EF-G with V67D ribosomes has no effect on $\tau_{unwinding}$ but lengthens $\tau_{release}$ (Supplementary Fig. 4a, c).

Taken together, the observed insensitivity of $\tau_{unwinding}$ to perturbed GTP hydrolysis and $P_i$ release clearly indicates that hairpin unwinding by the ribosome does not require GTP hydrolysis or $P_i$ release by EF-G. On the other hand, the sensitivity of $\tau_{release}$ indicates that GTP hydrolysis and $P_i$ release are required for EF-G release from the ribosome following productive translocation.

### Unproductive EF-G binding events do not require GTP hydrolysis

In the hairpin unwinding fleezers assay, EF-G binding is sometimes followed by unbinding without a hairpin unwinding step. These "unproductive" binding events occur randomly throughout the molecular trajectories of the ribosomes (Supplementary Fig. 6a), indicating that they represent a natural sampling behavior of competent translating ribosomes. The mean dwell time of unproductive events ($\tau_{unproductive}$) is 300 ± 80 ms for WT EF-G with GTP (Supplementary Fig. 6b, magenta dots, Supplementary Fig. 6c, top green bar). Interestingly, $\tau_{unproductive}$ is not lengthened by perturbed GTP

hydrolysis and $P_i$ release (Supplementary Fig. 6b, blue, green, and purple dots, Supplementary Fig. 6c, bottom three bars), unlike the total dwell time for productive events ($\tau_{unwinding}+\tau_{release}$). This insensitivity indicates that the release of EF-G in unproductive events, in contrast to that of productive ones, does not require GTP hydrolysis. In turn, this observation suggests that during a productive event, after an initial "loose" and reversible binding, EF-G switches into a "tight" binding mode (that will require GTP hydrolysis and $P_i$ release to unbind), whereas during unproductive events EF-G remains loosely bound and can unbind without GTP hydrolysis. Indeed, the existence of multiple modes of EF-G binding to the ribosome has been described by structural[18,22,23] and functional[35] studies. While we cannot rule out that the initial EF-G binding mode could be different in these two types of events, for example, by transient binding of inactive or misfolded EF-G in the case of the unproductive events, the existence of reversible binding of EF-G·GTP is supported by single-molecule and bulk measurements[8,36].

### Hairpin unwinding and forward 30S head rotation are temporally correlated

What is the conformational event responsible for unwinding of the mRNA hairpin during EF-G dwell on the ribosome? It has been previously shown that slowing down reverse 30S head rotation does not delay hairpin unwinding[32], indicating that unwinding occurs before this rotation. Could forward 30S head rotation be responsible for hairpin unwinding?

To address this question, we first assessed the temporal relationship between hairpin unwinding and forward or reverse head rotation by reconstituting ribosomes with Atto550 and Atto647N fluorophores attached to proteins S12 and S19 on the body and head domains of the 30S, respectively. FRET between these positions reports on the state of 30S head rotation (low FRET corresponding to the rotated state) as shown in bulk stopped-flow experiments[24,37] (Supplementary Fig. 7a, b). We performed the hairpin fleezers assay with non-labeled EF-G and doubly-labeled ribosomes to simultaneously monitor multi-turnover hairpin unwinding and single-molecule FRET (Fig. 2a). The trajectories reveal that in most cases, hairpin unwinding occurs around the time that a change from high to low FRET is observed, as expected if forward head rotation coincides with hairpin unwinding (Fig. 2b, c). Although shot noise does not allow the precise timing of FRET change for individual steps, event averaging ($n = 396$) around the unwinding time clearly shows the FRET change to correlate with unwinding (Fig. 2d). Average FRET starts dropping a few milliseconds before unwinding, reaches a minimum around the time of unwinding, and then slowly recovers over a few hundred milliseconds. Fitting to a two-sided exponential yields rate estimates for forward ($115 \pm 30\,s^{-1}$) and reverse ($5.3 \pm 0.4\,s^{-1}$) head rotation, in general agreement with bulk results[24,36,37] (Supplementary Fig. 7c, d).

The observed temporal correlation under unperturbed conditions raises the possibility that hairpin unwinding results from forward head rotation. However, as shown next, such a causal relationship likely does not exist.

### Hairpin unwinding does not result from mRNA translocation relative to the 30S body domain

To better estimate when hairpin opening occurs relative to the movement of mRNA with respect to the 30S body (which likely corresponds to forward head rotation), we implemented an alternative fleezers assay that monitors when the movement of mRNA relative to the 30S body domain occurs with respect to EF-G binding. In this assay, DNA handles are used to tether the 5' end of a hairpin-less mRNA to one bead and the biotinylated protein S16 of the ribosome[7] to the other bead (Fig. 3a, d). The applied tension in this geometry assists ribosome translocation along the mRNA, and movement of mRNA relative to the 30S body in each translocation cycle increases the tether extension by one codon (3 nt). We obtained translocation traces in this novel assisting-force geometry and simultaneously monitored Cy3-labeled EF-G binding (Fig. 3g, Supplementary Fig. 8a). We refer to the time between EF-G binding and mRNA stepping (relative to the 30S body) in this assay as $\tau_{pre}$, and the time between the step and EF-G unbinding as $\tau_{post}$ to distinguish them from $\tau_{unwinding}$ and $\tau_{release}$ in the hairpin unwinding assay (Fig. 3b, c, e, f).

In the presence of WT EF-G and GTP, the mean total dwell times of EF-G in the assisting force assay are similar to those observed in the hairpin unwinding assay ($\tau_{unwinding} + \tau_{release} = 445 \pm 70$ ms, $\tau_{pre} + \tau_{post} = 308 \pm 55$ ms, Supplementary Fig. 9, top row). Furthermore, $\tau_{pre}$ is marginally longer than $\tau_{unwinding}$ ($\tau_{pre} = 151 \pm 45$ ms, $\tau_{unwinding} = 72 \pm 14$, Fig. 3h, top row, left cyan and green bars), suggesting that hairpin unwinding coincides with or is shortly followed by mRNA-body translocation under unperturbed conditions. In the presence of 0.2 mM fusidic acid, an antibiotic that has been shown to slow down only late stages of translocation including reverse 30S head rotation[23,38], we observed lengthening of $\tau_{post}$ but not $\tau_{pre}$ (Fig. 3h, second row, Supplementary Fig. 8b), consistent with the idea that tether extension change in the assisting force assay corresponds to forward, and not reverse, head rotation.

Similar to the hairpin assay, we subjected ribosomes in the assisting force assay to conditions that impede GTP hydrolysis and/or $P_i$ release. Notably, under all of the four perturbed conditions tested, we observed lengthening of $\tau_{pre}$ and, more severely, $\tau_{post}$ on the order of seconds (Fig. 3h, bottom four rows, Supplementary Fig. 8c-f): in the presence of WT EF-G with 0.25 mM GTP mixed with an excess (0.5 mM) of either GDPNP or GTPγS, or in the presence of H92Q EF-G mutant with 1 mM GTP, $\tau_{pre}$ was lengthened to 1–3 s and $\tau_{post}$ was lengthened to 2–7 s. More dramatically, the use of H92A EF-G with 1 mM GTP lengthened $\tau_{pre}$ and $\tau_{post}$ to more than 10 s and 20 s, respectively. The ribosome trajectories in this assay exhibit forward and backward steps in rapid succession during $\tau_{pre}$ before displaying a decisive step forward (Supplementary Fig. 8c-f), suggesting reversible mRNA stepping attempts before eventual irreversible mRNA translocation relative to the 30S body occurs. Mean total EF-G dwell times ($\tau_{pre} + \tau_{post}$) in the assisting force assay under GTPase perturbed conditions are longer compared to those in the hairpin unwinding assay ($\tau_{unwinding} + \tau_{release}$), likely due to differences in experimental conditions including the analog concentration ratios used (Supplementary Fig. 9, bottom four rows).

The assisting force assay shows that the time between EF-G binding and mRNA translocation relative to the 30S body ($\tau_{pre}$) is sensitive to the use of EF-G mutants and GTP analogs, whereas the hairpin assay shows that hairpin unwinding time ($\tau_{unwinding}$) is insensitive to those perturbations. Together, these results strongly suggest that hairpin unwinding and mRNA translocation relative to the 30S body are not the same event despite their temporal correlation under normal conditions, and that they can be uncoupled such that unwinding precedes mRNA-body translocation when using EF-G mutants and GTP analogs. The observation that mRNA-body translocation is delayed under GTPase perturbations and hairpin unwinding is not, indicates that translocation is not responsible for driving hairpin unwinding.

### mRNA translocation relative to the 30S body is accelerated by GTP hydrolysis but does not require it

Consistent with the established notion that GTP hydrolysis accelerates mRNA translocation, the assisting-force assay results (Fig. 3h) show that both $\tau_{pre}$ and $\tau_{post}$ are lengthened by GTPase perturbations. In particular, this observation mirrors previous bulk single-turnover studies[24,36] in which GTPase perturbations delayed both forward and reverse 30S head rotations, corresponding to mRNA-body translocation and EF-G release here, respectively.

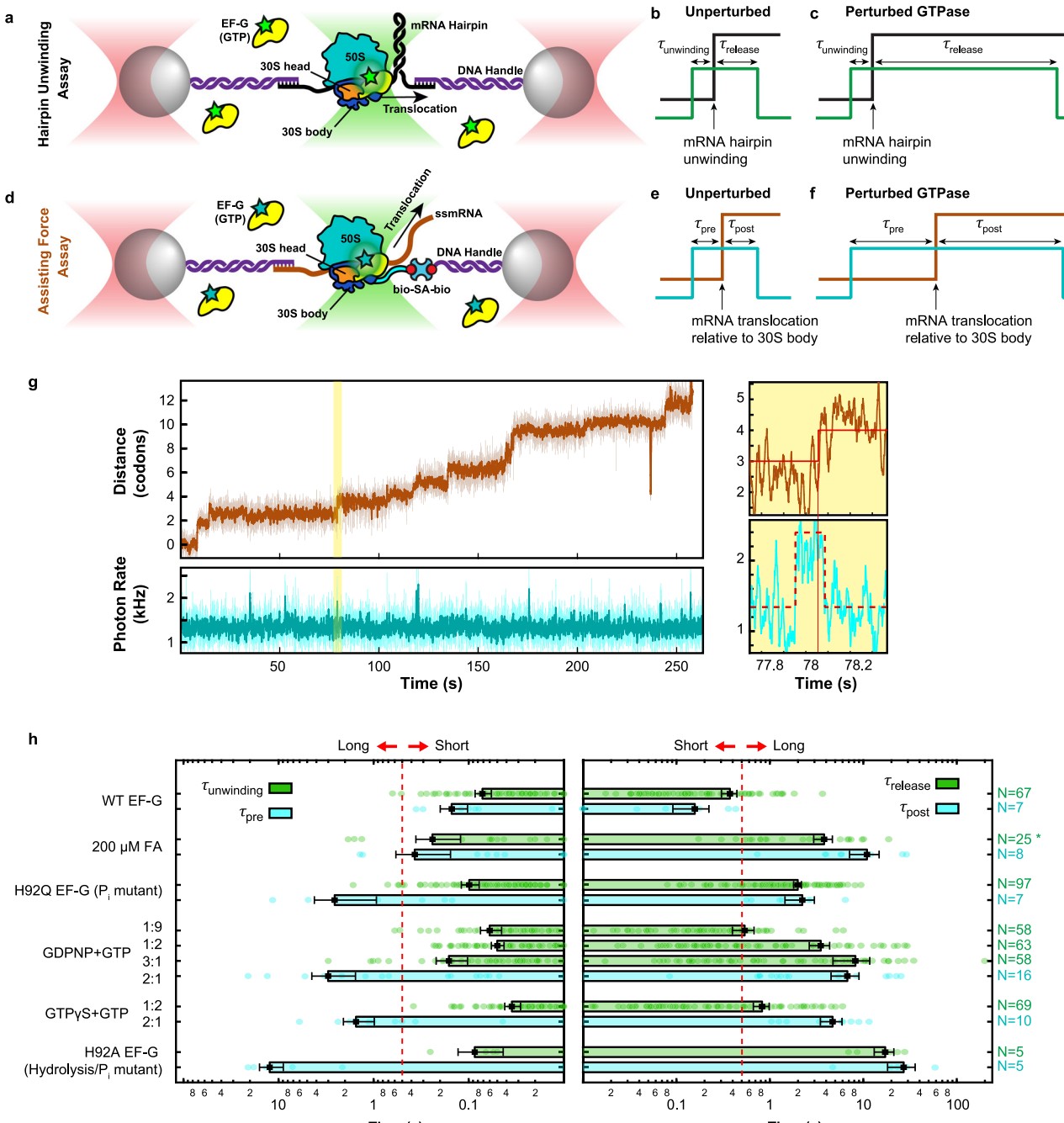

**Fig. 3 | Dependence of mRNA translocation relative to the 30S body on GTP hydrolysis and $P_i$ release.** **a**–**c** Schematics of the experimental setup (**a**) and readout under unperturbed (**b**) and perturbed GTPase conditions (**c**) for the hairpin unwinding fleezers assay, shown for comparison. **d**–**f** Schematics of the experimental setup (**d**) and readout under unperturbed (**e**) and perturbed GTPase conditions (**f**) for the assisting force fleezers assay for simultaneous detection of mRNA translocation relative to the 30S body and EF-G binding during ribosomal translocation. **g** A fleezers trajectory of consecutive steps taken by a single ribosome in the presence of wild-type EF-G and GTP. The shaded area is magnified in the right panels in which the detected transitions are demarcated by red lines. **h** Summary of measurements from the hairpin unwinding assay (green bars) and the assisting force assay (cyan bars), represented as mean ± standard error. Individual data points are shown as small circles. The $\tau_{pre}$ and $\tau_{unwinding}$ measurements are shown in the left panel, and $\tau_{post}$ and $\tau_{release}$ measurements are shown in the right panel for various experimental conditions. The conditions are listed on the left and the number of data points for each condition (N) are shown on the right. Note that the time axis is shown in logarithmic scale to better separate short (normal) and long times, as demarcated by the dashed red lines. For GDPNP and GTPγS, a higher analog:GTP ratio was used in the assisting force assay (2:1) compared to the hairpin unwinding assay (1:2). Fusidic acid (FA) measurements in the hairpin assay (*) were made previously[32]. Source data are provided as a Source Data file.

The lengthening of both $\tau_{pre}$ and $\tau_{post}$ by GTPase perturbations can be rationalized by a model in which GTP hydrolysis, which normally occurs rapidly upon ribosome-EF-G binding, and possibly $P_i$ release, kinetically accelerate mRNA-body translocation without being thermodynamically required for it, while dissociation of EF-G from the ribosome requires GTP hydrolysis and $P_i$ release. In this model, when GTP hydrolysis is perturbed or even abolished by the use of analogs or mutants, mRNA-body translocation would still occur, albeit very slowly (i.e., not catalyzed), resulting in the observed lengthening of $\tau_{pre}$. Importantly, this model also predicts that $\tau_{post}$ will be lengthened by

the perturbations because EF-G dissociation requires GTP hydrolysis and $P_i$ release, and under these conditions, these processes could occur after mRNA-body translocation. Alternative models in which GTP hydrolysis is strictly required for mRNA-body translocation predict a lengthened $\tau_{pre}$, but since in those models GTP hydrolysis has already occurred by the time of mRNA-body translocation, they cannot satisfactorily explain why the subsequent dissociation of EF-G ($\tau_{post}$) is also delayed. Predictions of each model for $\tau_{pre}$ and $\tau_{post}$ under GTPase perturbations are described in Supplementary Fig. 10. Our observations in the assisting force assay best match the model in which GTP hydrolysis and $P_i$ release catalyze but do not thermodynamically fuel, and are not required for, mRNA translocation relative to the 30S body.

## Discussion

Unwinding of an mRNA hairpin is an endergonic process that requires free energy input. Similarly, unidirectional mRNA translocation relative to the 30S body domain in precise codon steps is also endergonic, since in the absence of an energy source, entropy favors bidirectional movement with variable step sizes. Our results show that the ribosome-EF-G complex carries out these processes without requiring EF-G-mediated GTP hydrolysis or $P_i$ release, therefore excluding these GTPase reactions as necessary direct free energy sources for them to occur. We propose that the ribosome derives the energy required for unwinding and mRNA translocation relative to the 30S body from its binding to EF-G. Interestingly, we have found that the initial binding of EF-G precedes unwinding by ~70 ms on average ($\tau_{unwinding}$). We thus hypothesize that after the initial EF-G binding, a transition from a "loose" to a "tight" binding mode supplies the free energy that facilitates conformational changes leading to hairpin opening and mRNA translocation relative to the 30S body. This interpretation is supported by the observation of unproductive EF-G binding events.

Our findings enable us to propose a model for the ribosomal translocation pathway. By comparing hairpin unwinding and assisting force assay results under perturbed conditions (Fig. 4a), we can infer the existence of four intermediates (states I to IV) during EF-G dwell on the ribosome, resulting from EF-G binding, hairpin unwinding, mRNA translocation relative to the 30S body, and $P_i$ release, respectively. GTP hydrolysis has no thermodynamic or kinetic effect on hairpin unwinding; it can kinetically accelerate mRNA translocation relative to the 30S body but is not thermodynamically required for this step either. State IV (GDP-bound conformation) is the first state to thermodynamically require GTP hydrolysis and $P_i$ release. This order of states under perturbed conditions reflects causal rather than temporal precedence. $P_i$ release may temporally precede mRNA-body translocation when not delayed (Fig. 4a, bottom), and we do not rule out a branched pathway in which Pi release precedes mRNA-body translocation in one path and follows it in the other[15].

Previous observations of rapid GTP hydrolysis by EF-G combined with results of kinetic fluorescence measurements with EF-G mutants and GTP analogs have led to a GTP hydrolysis-driven power-stroke model for translocation[4,15,17,18,35,39,40]. This model posits that EF-G conformational changes, fueled by the energy of GTP hydrolysis, drive the mechanical movement of mRNA and tRNAs relative to the 30S body domain. However, the fact that hydrolysis normally occurs before an event during translocation does not necessarily imply that it is energetically required for the event. GTP hydrolysis presumably precedes both hairpin unwinding and mRNA-body translocation, but our results indicate that it is not required thermodynamically to drive either of these endergonic events. Importantly, even if an event such as mRNA-body movement is accelerated as a result of GTPase activity, it is not necessarily the case that the energy of GTP hydrolysis is used in that acceleration. Rather, EF-G (after hydrolysis) can act as a catalyst, and acceleration will result simply from its preferential binding to, and stabilization of the transition state immediately preceding the event[41]. Clearly, the energy of this binding is unrelated to any energy released

up to this event by hydrolysis. Thus, the kinetic acceleration observed in the presence of GTP hydrolysis does not imply that it utilized the energy resulting from hydrolysis. Our results are in agreement with early studies suggesting that GTP hydrolysis is required for EF-G dissociation[19], with single-molecule FRET measurements showing that events preceding deacyl-tRNA release from the ribosome are not dependent on GTP hydrolysis[42], and with bulk FRET measurements showing that blocking GTP hydrolysis abolishes only the reverse but not the forward 30S head rotation[24]. The recent discovery of an EF-G paralog that catalyzes translocation without GTP hydrolysis[43] further argues against a driving role of EF-G mediated GTP hydrolysis in mechanical movements during ribosomal translocation.

The energy expenditure for EF-G-catalyzed translocation (including the endergonic unwinding of mRNA secondary structures; 1–9 kcal/mol/codon for hairpins[44]) must be ultimately supplied in excess by exergonic reactions that are coupled to translocation, including peptidyl transfer provided by aminoacyl-tRNAs (3–4 kcal/mol[7]), and GTP hydrolysis and $P_i$ release provided by EF-G (10-12 kcal/mol[45]). We propose that initially, other exergonic events such as EF-G tight binding and conformational changes of the ribosome triggered by this binding energetically drive mRNA unwinding and mRNA-body translocation. However, this expenditure of binding or conformational energy to perform work also traps the EF-G-ribosome complex in a low energy state. To release EF-G from this trapped state and reset the ribosome conformation, the complex must be "rescued" by an external source of energy provided in this case by GTP hydrolysis and $P_i$ release, or by the energy stored in the complex from these events if they occur earlier. Interestingly, the implied sub-micromolar affinity of EF-G·GTP for the ribosome based on $K_m$ measurements[34] corresponds to 9-10 kcal/mol of binding energy, sufficient for unwinding of stable hairpins yet still rescuable by GTP hydrolysis.

Accordingly, a conceivable energy landscape for translocation is shown in Fig. 4b. In the presence of EF-G, progression up to state III (mRNA-body translocation) is downhill largely due to the tight binding of EF-G. It is roughly similar with or without GTP hydrolysis, although the kinetic barrier to state III is lowered with (catalyzed by) hydrolysis (Fig. 4b, red and green traces). Progression through the remainder of the cycle is favored by the exergonic $P_i$ release (red trace after state III in Fig. 4b), which energetically pays for EF-G release. EF-G unbinds from the ribosome after $P_i$ release, and reverse head rotation takes place to reach the post-translocation state. If GTP is not hydrolyzed or $P_i$ is not released, completion of the cycle will be energetically uphill, requiring release of the tightly bound EF-G from the ribosome (green trace after state III in Fig. 4b). In the absence of EF-G, the elongation cycle is driven only by the free energy supplied by peptidyl transfer (on average, notwithstanding codon differences), and its translocation phase alone is likely even less favorable and navigates a rough energy landscape that significantly limits its overall rate (Fig. 4b, black trace). Attainment of the post-translocation state and the ensuing multi-turnover translocation is still possible in this condition, although only at a very slow rate[30,31] and with compromised frame maintenance[46,47].

It remains unknown what structural change in the ribosome-EF-G complex leads to unwinding of the mRNA hairpin in state II. Since the optical tweezers pulling force alone is insufficient to unwind the codon duplex without the ribosome's assistance[2,32,48], an exergonic change of state in the ribosome complex upon EF-G binding must be involved, which may include binding of the unwound mRNA to the ribosome[49]. We show that mRNA-body translocation (likely corresponding to complete forward 30S head rotation) is not responsible for driving hairpin unwinding. Nevertheless, unwinding may still be in some way linked to head rotation; for example, it could be triggered by a partial (4°) head rotation that precedes full (20°) rotation and moves the mRNA by ~1 nt[18,22,23]. The partial rotation is associated with a dynamic

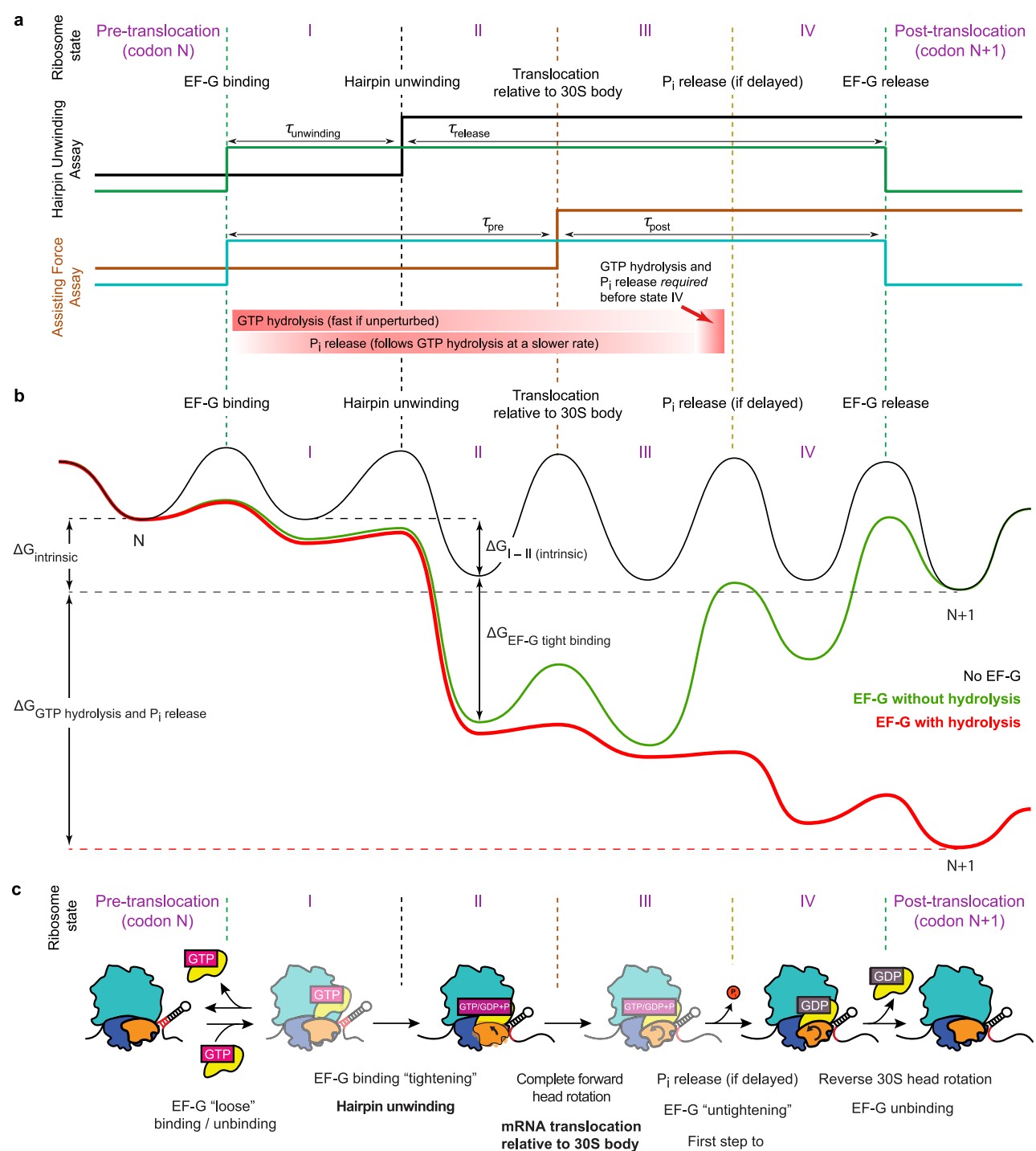

**Fig. 4 | Model for causal and energetic requirements of translocation. a** A model for ribosomal translocation mapped onto the signature of observables from the hairpin unwinding and assisting force fleezers assays. GTP hydrolysis and $P_i$ release are required neither for hairpin unwinding nor for mRNA translocation relative to the 30S body, although they can kinetically accelerate the latter (attainment of state III). They are rather needed for the subsequent release of EF-G·GDP to reset the ribosome before the next translocation cycle. Note that under normal conditions, hydrolysis is fast and Pi release can occur before or after mRNA-body translocation (bottom, red gradients), but they are strictly required only to progress to state IV which is followed by EF-G dissociation (red arrow). **b** A qualitative model, consistent with experimental results, of the energy landscape of translocation under normal conditions (in red) compared with non-canonical conditions (without GTP hydrolysis or without EF-G altogether, in green and black, respectively). The three energy landscapes do not share identical intermediate states but are depicted with roughly corresponding ribosomal configurations for visual clarity. According to this model, EF-G binding energy and the energy stored in intrinsic ribosomal conformations drive the early steps of translocation. Only the general features of the energy landscape, not the exact heights of free energy basins and barriers, are proposed here as described in the text. **c** A speculative structural assignment of translocation intermediates under GTPase perturbation. Ribosome-EF-G complexes with partial and full 30S head rotations observed in time-resolved structures[18,22,23] are assigned to states II and IV, respectively. Confidence in structural assignment for states I and III (shown in fainter colors) is weaker; these states could correspond to EF-G recruitment by the ribosomal L7/L12 stalk[57], and head-rotated ribosome-EF-G complexes trapped with non-hydrolyzable GTP analogs[58–60], respectively. The structural features of each state are depicted in the cartoons, and changes at each transition are described at the bottom.

state of translocation identified as INT1 previously[16,23]. A speculative assignment of states I to IV to currently available structures can be made based on this possibility (Fig. 4c). However, further studies will be necessary to structurally characterize these states.

To summarize, our study indicates that the free energy from EF-G·GTP binding to the ribosome initially drives ribosomal translocation, that GTP hydrolysis catalyzes but is not thermodynamically required for mRNA-body translocation, and that $P_i$ release pays for the dissociation of EF-G and the resetting of the ribosome at the end of each translocation cycle.

## Methods

### Buffers and reagents

**Synthesis of DNA handles.** DNA handles (2.5 kb) were generated by PCR using a forward primer modified to contain a 5′ biotin (for bead attachment in hairpin unwinding assays) or digoxigenin (assisting force assays), and a reverse primer containing either an internal 3-carbon spacer (iSpC3) to generate a 21-nt 5′ overhang (for mRNA 5′ annealing), a string of inverted bases to generate a 22-nt 3′ overhang (for mRNA 3′ annealing), or a 5′ biotin (to attach to ribosomes in assisting force assays). PCR products were cleaned up (Qiagen). The digoxigenin-biotin handle was pre-incubated with 10x excess of free streptavidin (Sigma) prior to deposition.

**Preparation of mRNA and tRNA.** The V50 mRNA which encodes 50 valine codons in the 5′ strand of its hairpin[32] (for unwinding assays) and a hairpin-less version of it lacking the 3′ strand of the hairpin (for assisting force assays) were synthesized from EcoRI-linearized plasmid templates using MEGAscript T7 Transcription Kit (Invitrogen) at 37 °C for 4 h, purified by polyacrylamide gel electrophoresis separation followed by phenol-chloroform extraction, ethanol precipitation, and G25 desalting, and stored at −80 °C. Total tRNA (from Escherichia coli MRE 600, Roche) was deacylated by heat treatment and purified via phenol-chloroform extraction and ethanol precipitation[32]. For experiments with total tRNA, charging was performed at the time of translation mix preparation (see below). For experiments with valine tRNA only, 10 units of valine-specific tRNA (Sigma) were charged using DEAE-purified S100 extract[32].

**Preparation and labeling of ribosomes and proteins.** The preparation of E. coli MRE600 tight-couple 70S ribosomes[50], S16-biotinylated ribosomes[7], and DEAE-purified S-100 extracts[51] was performed as previously described. FRET reporter ribosomes were prepared by in vitro reconstitution[24] using fluorophores Atto550 and Atto647N (Attotec). For V67D L7/L12 ribosomes, MRE600 50S subunits were depleted of L12 by $NH_4Cl$/Ethanol treatment[34], reconstituted with mutant L12 by incubating at 37 °C for 30 min, and associated with natural 30S subunits as described[52]. Histidine-tagged IF-1, IF-2, IF-3, Val-tRNA synthetase, EF-Tu, and EF-G (unlabeled) were prepared as described[32]. For labeled EF-G, the single-cysteine variant EF-G(S73C)[8] was purified and labeled with Cy3 as described[32]. The EF-G mutants (H92Q and H92A) were generated from this variant using site-directed mutagenesis (QuickChange, Agilent) and labeled in the same way. The V67D L12 mutant was generated by site-directed mutagenesis from the pSV281 plasmid encoding L12[53] and cloned into pET21b excluding any tags. The untagged mutant protein was then expressed in E. coli BLR (DE3) cells essentially as described[54] and purified by ion-exchange and size exclusion chromatography using Resource Q, Superdex75, and Resource S columns (Pharmacia Biotech). All proteins were stored in 25 mM Tris·HCl pH 7.5, 60 mM $NH_4Cl$, 10 mM $MgCl_2$ and 5 mM βME at −80 °C.

**Preparation and deposition of stalled ribosome complexes.** Ribosome initiation complex formation was carried out as described previously[32] for natural, biotinylated, L7/12 mutant (V67D), or FRET ribosomes, except that a longer incubation time (30–40 min) was used for the FRET ribosomes to improve the yield. Subsequent translocation and stalling of ribosomes at the Lys9 codon was also performed in bulk as described[32] to select for active ribosomes and to allow annealing of the 5′ handle (containing biotin or digoxigenin for the unwinding and assisting force assays, respectively). Stalled complexes were flash-frozen and stored at −80 °C. For the unwinding assays, a 2 μL stalled complex aliquot was mixed with 1 μL of a 0.1% suspension of 1 μm streptavidin-coated polystyrene beads (Bangs Labs); separately, 1 μL of 100 nM 3′ overhang handle was mixed with 1 μL of the same beads. For assisting force assays, a 2 μL stalled complex aliquot was mixed with 1 μL of a 0.1% suspension of 1 μm anti-digoxigenin antibody-coated beads[55], and separately, 1 μL of 100 nM biotin-digoxigenin handle (pre-incubated with streptavidin) was mixed with 1 μL of the same beads. After deposition by incubation at room temperature for 10–20 min, the beads were suspended in 1 mL of 1x TLC buffer (40 mM HEPES pH 7.5, 60 mM $NH_4Cl$, 10 mM Mg(OAc)$_2$, 6 mM βME) and injected in separate channels of the fluidic chamber.

**Optical tweezers measurements.** Tethers were formed inside the fluidic chamber by placing a stalled-ribosome bead in one optical trap and a handle bead in another. Tethering was made through hybridization of mRNA and the 3′ handle in the unwinding assays, and by biotin-streptavidin binding in the assisting force assay. For unwinding assays, correct tethers containing a single stalled ribosome-mRNA complex were identified by their force-extension unfolding signature[32], and for assisting force assays, correct tethers were simply identified by their contour length obtained from force-extension curves. The translation mix was then delivered via a shunt to the tethered stalled ribosomes, while the tether was held at relatively high forces (13.5–14.5 pN for unwinding assays and 9–12 pN for assisting force assays) by manually adjusting the trap distances as needed. These forces were chosen to be sufficiently high to provide single-codon resolution, and in the case of the unwinding assay, also far below the critical force[2,32,48] of the hairpin so that hairpin unwinding would not occur spontaneously without the participation of the active ribosome (Supplementary Fig. 1c).

**Translation mix.** Typically, the translation mix (250 μL) consisted of GTP (Promega), GDPNP (Sigma), and GTPγS (Roche) at concentrations indicated in the text, 1 mM ATP, 420 μM valine, 12 mM creatine phosphate (CP), 0.4 U/μL RNaseOut (Invitrogen), 2 μM valine tRNA (in total tRNA), 3.9 μg/mL creatine kinase (CK), 3 μg/mL myokinase (MK), 0.08 μM nucleoside diphosphate kinase (NDPK), 1 μg/mL pyrophosphatase (PP), 1 μM val-tRNA synthetase, 4 μM EF-Tu, and either 10 nM Cy3-labeled EF-G (WT or a mutant as indicated) or 1 μM unlabeled EF-G, in 1x TLC buffer augmented with an oxygen scavenging system (see below) to improve tether and fluorophore lifetimes. The CP, CK, MK, NDP, and PP constitute a phosphate regeneration system. For assisting force assays, a simpler translation mix (250 μL) was used that lacked the regeneration system and only contained 2 μM EF-Tu·GTP·val-tRNA$^{val}$ ternary complex pre-formed as described previously[49] at 50 μM in 10 μL, plus EF-G, GTP (or analogs), and the buffer with oxygen scavenging system.

For experiments with labeled EF-G, the oxygen scavenging system consisted of 2 mM Trolox, 0.8% glucose, 0.3 mg/mL glucose oxidase (Sigma), 0.04 mg/mL catalase (Sigma), and 0.5 U/μL RNaseOut. For FRET experiments, the mix contained 2 mM Trolox, 0.8% glucose, 0.6 mg/mL glucose oxidase, 0.2 mg/mL catalase, and 1 U/μL RNaseOut. Additionally, for FRET experiments, the HEPES concentration was raised to 200 mM for a stronger buffer, and βME was omitted to improve acceptor fluorophore behavior. Due to the acidification of the translation mix by the oxygen scavenging system, the mix was made fresh every 4 h. The mix was passed through a 0.22 μm filter prior to addition to the optical tweezers chamber.

**Extension and fluorescence measurements.** Extension and fluorescence data were typically collected at 1.33 kHz and 1 kHz, respectively. The instrument was controlled using LabView and the data were visualized and analyzed in MATLAB[56]. Extension was converted to contour length (and number of codons) using extensible worm-like chain parameters extracted from tether force-extension curves. For experiments with labeled EF-G, transition times for extension (steps) were detected using Hidden Markov Model (HMM) fitting, and those for fluorescence (EF-G binding events) were detected using a combination of HMM and Pruned Exact Linear Time fitting[32]. Only well-resolved single steps that occurred during well-resolved binding events were included for analysis of productive events, and only well-resolved binding events that did not overlap with steps were included for analysis of unproductive events. The observation of very long (>40 s) fluorescence lifetimes for some of the conditions tested, e.g. the EF-G(H92A) mutant or GDPNP used at high ratios, demonstrated that fluorophore lifetimes do not limit the validity of typical observed event times (a few seconds or less) in these assays. For FRET experiments, FRET ratio was calculated after applying low-pass and median filters to normalized background-corrected donor and acceptor fluorescence signals. A contrast-adjusted "false color" plot, in which the red and green color values correspond to FRET and 1-FRET, was made for better visualization. For event averaging, multiple events were aligned by their unwinding time, and their FRET ratios and extension changes were median-averaged. For estimation of forward and reverse head rotation rates ($k_f$ and $k_r$), a two-sided five-parameter exponential function,

$$\text{FRET}(t) = \begin{cases} h - (h-l)e^{k_f(t-m)}, \, t < m \\ h - (h-l)e^{k_r(m-t)}, \, t \geq m \end{cases} \quad (1)$$

was fitted to the FRET average using the least squares method, where $h$ and $l$ are the high and low average FRET values, respectively, and $m$ is the center time (Supplementary Fig. 7c).

**Fitting of distributions.** For conditions with higher throughput, fitting of EF-G dwell time was done using the maximum likelihood estimation (MLE) method, and characteristic time estimates and 95% confidence intervals are reported. The distribution of $\tau_{\text{release}}$ for H92Q fit well to a single exponential distribution,

$$P_s(t) = k_{single}e^{-k_{single}t} \quad (2)$$

with the characteristic time $<\tau> = 1/k_{single}$. The distribution of $\tau_{\text{release}}$ for all other conditions tested was best fit to a bi-exponential distribution,

$$P_{bi}(t) = f_1 k_1 e^{-k_1 t} + (1 - f_1)k_2 e^{-k_2 t} \quad (3)$$

with individual characteristic times $<\tau_1> = 1/k_1$ and $<\tau_2> = 1/k_2$, and the overall characteristic time $<\tau> = f_1/k_1 + (1-f_1)/k_2$ where reported. The distribution of $\tau_{\text{unwinding}}$ was fit to a bi-exponential distribution (Eq. 3) for all conditions tested. Note that for experiments performed with a mixture of GTP and GTP analogs, $f_1$ for $\tau_{\text{release}}$ represents the slow population likely associated with the analog. Results for fits for $\tau_{\text{unwinding}}$ are shown in Supplemental Table 1, results for fits for $\tau_{\text{release}}$ are shown in Supplemental Table 2, and results for fits for $\tau_{\text{unproductive}}$ are shown in Supplemental Table 3. The evaluation of statistical significance for different conditions, compared to the WT condition, is shown in Supplemental Table 4. The tests were chosen to assess significance despite the limited sample sizes obtained. In the assisting force assay, compared to the unwinding assay, the stepping signal is smaller (3 versus 6 nt), and the lower signal is exacerbated by noise from the mechanical dynamics arising from ribosome tethering in this assay. These limit the number of data points that can be reliably analyzed for precise time measurements and further compromise the

throughput. For all conditions with lower throughput (Supplemental Table 5), or when comparing hairpin unwinding and assisting force assay results (in Fig. 3 and Supplementary Fig. 9), sample means and their standard errors are used instead of fit values for data analysis and visualization.

## Statistics and reproducibility

All data sets were reproducible under repeated rounds of data collection. Data from all rounds under each condition were combined to arrive at the final set for that condition. Data for different conditions were collected at a random order and were interleaved to minimize systematic errors. No statistical method was used to predetermine sample size. Sample sizes were sufficient for demonstrating statistically significant differences between GTPase-perturbation groups and the control group (WT EF-G with GTP) as shown in Figs. 1 and 3 and by the statistical tests shown in Supplementary Table 4. Some data points were excluded based on pre-established criteria, including a low signal-to-noise ratio in the optical tweezers or the fluorescence channel, or the presence of obvious experimental artifacts that precluded reliable measurements. No blinding was implemented. However, data from all groups were analyzed via the same pre-defined procedure.

## Reporting summary

Further information on research design is available in the Nature Portfolio Reporting Summary linked to this article.

## Data availability

The data supporting the findings of this study are available from the corresponding authors upon request. Source data for the figures and Supplementary Figs. are provided as a Source data file.

## Code availability

The codes used to view and analyze optical tweezers data are available from the corresponding author upon request.

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

## Acknowledgements

We thank Clint Spiegel (Western Washington University) for providing the plasmid encoding L12. We are grateful to Jamie Cate (University of California, Berkeley), Andrei Korostelev (University of Massachusetts Chan Medical School), Pohl Milon (Universidad Peruana de Ciencias Aplicadas), and Scott Blanchard (St. Jude Children's Hospital) for critical feedback on this manuscript, and to current and former members of the Bustamante laboratory for useful discussions. This work was funded by Nanomachine program KC1203, the Office of Basic Energy Sciences of the U.S. Department of Energy, contract no. DE-AC02-05CH11231 (C.B.), Howard Hughes Medical Institute (C.B.), National Institutes of Health National Institute of General Medical Sciences R01GM071552 (C.B.), National Institutes of Health National Institute of General Medical Sciences R01GM032543 (C.B.), National Institutes of Health National Institute of General Medical Sciences R35-GM118156 (H.F.N.), and Pre-doctoral Ford Foundation Fellowship (W.J.V.).

## Author contributions

Conceptualization: H.A., W.J.V., V.P.D., H.F.N., C.B. Investigation: H.A., W.J.V., G.R., B.S., L.L. Writing: H.A., W.J.V., C.B. Both H.A. and W.J.V. have the right to list their name first in their respective CVs.

## Competing interests

The authors declare no competing interests.
