## [Transparent Peer Review file · Nature Communications]

The Ribosome Derives the Energy to Translocate and Unwind mRNA from EF-G Binding

Corresponding Author: Professor Carlos Bustamante

Version 0:

Reviewer comments:

Reviewer #1

(Remarks to the Author)

The paper by Van Patten et al. presents experiments that simultaneously monitor several steps of tRNA–mRNA translocation on the ribosome using high-resolution optical tweezers with fluorescence detection, a method validated in their previous work. This technique enables the reconstruction of the order of events, including mRNA unwinding relative to EF-G binding, mRNA unwinding versus 30S head rotation, and mRNA translocation versus EF-G binding.

While the experiments appear robust overall, aside from some technical issues noted below, the authors do not fully address the questions they initially pose. For example, they begin by asking about the GTP requirement for translocation but then discuss their data in terms of the ground-state thermodynamics of tRNA movement—specifically, whether the energy from GTP hydrolysis directly drives translocation by stabilizing the ground state of reaction intermediates. The distinction between the “mere acceleration” of translocation and a “requirement” for GTP hydrolysis is difficult to follow, as translocation can proceed spontaneously, even in the absence of GTP or EF-G. Thus, it is unclear why “requirement” is interpreted in terms of ground-state stabilization while “acceleration” is (correctly) associated with lowering the transition state barrier.

The literature referenced in the manuscript consistently indicates that GTP hydrolysis and subsequent Pi release reduce the transition state barrier for mRNA translocation and promote EF-G release from the ribosome. Earlier kinetic studies suggested that Pi release, rather than GTP hydrolysis per se, is the key step, and recent cryo-EM structures (refs. 18 and 22) demonstrate how Pi release is coupled to tRNA translocation. Most models interpret the effects of Pi release in terms of a reduced transition state barrier, with little evidence supporting the idea that GTP hydrolysis stabilizes or destabilizes the ground state or “drives energetically uphill events” (c.f. p. 2). Power-stroke models, which the authors mention, focus on rate acceleration and, additionally, propose that conformational changes following Pi release prevent backward tRNA movement. There is no doubt that EF-G binding plays an important role in translocation, but the contribution of GTP hydrolysis and Pi release is essential, as they bring the rate of translocation to physiologically relevant levels. When these literature findings are compared with the data presented in Fig. 3—where replacing GTP with a non-hydrolysable analog increases both $\tau(\text{Pre})$ (the time to translocation) and $\tau(\text{Release and Post})$ (due to impaired EF-G release)—there is no apparent contradiction. Instead, the results are entirely consistent with established models. This section would benefit from revision to clarify this point and avoid suggesting inconsistencies where none exist.

A second concern relates to the authors' claims regarding the role of GTP hydrolysis and Pi release in stabilizing the ground state of the post-translocation complex. The paper does not provide direct evidence for this statement, as it relies solely on rate measurements, making it difficult to distinguish between ground-state stabilization/destabilization and changes in the transition state barrier. It is well established that without GTP hydrolysis, EF-G forms a stable complex with the ribosome, however this could reflect a high transition state barrier for EF-G dissociation rather than a true ground state stabilization. Additionally, it is well established that many tRNA pairs on the ribosome exhibit a favorable thermodynamic gradient for forward translocation. However, whether the tRNA(Val) used in this study shares this property remains unclear, and the authors may wish to experimentally assess whether the forward direction is indeed thermodynamically favorable.

There is also a potential issue with the energy profile presented in Fig. 4. The model begins with a pre-translocation complex (two tRNAs in the A and P sites) and ends with a complex containing only a tRNA in the P site—a state expected to be destabilized due to the loss of A-site interactions. This raises the question of why the final state (N+1) appears to have a lower energy than the pre-translocation state (N) after EF-G dissociation in the presence of GTP. A more detailed discussion of this aspect would improve clarity. Additionally, performing single-codon translocation experiments with tRNAs that thermodynamically favor backward translocation could provide further insight into the role of GTP hydrolysis.

The section on mRNA helix unwinding is also somewhat unclear. The data suggest that unwinding occurs early and is independent of both GTP hydrolysis and Pi release, aligning with the authors' previous work in *Molecular Cell*. However, the

overlap with their earlier findings and the novelty of these results are not clearly delineated, and the final mechanistic interpretation of unwinding remains somewhat speculative. Similarly, the observation that EF-G can bind and unbind from the ribosome before productive engagement has been reported in several smFRET studies and is already incorporated into most translocation models. Its inclusion in this study appears to serve completeness rather than introduce novel mechanistic insight.

In addition, the terminology used to describe conformational changes is confusing. The authors refer to “30S head rotation” without clarifying whether this term is equivalent to “swiveling” or how it relates to the formation of the chimeric state—a key intermediate in translocation, as shown by the Noller group. Cryo-EM studies (refs. 18 and 22) indicate that tRNA movement into the chimeric state requires Pi release. Notably, there is no disagreement between these two papers; rather, both differ from ref. 23 (as discussed in refs. 18 and 22). Given these findings, the order of events in Fig. 4 (where complete forward head rotation is shown before Pi release) appears to be reversed. Since Pi release is not measured directly in this study, the rationale for this ordering remains unclear. The discussion also relies on findings with the EF-G H92Q mutant, which is based on prior work of a different lab suggesting that this mutant retains GTPase activity but has reduced Pi release (ref. 33). However, concerns arise from the fact that the supposedly GTPase-deficient H92A mutant still shows significant GTPase activity (ref 33), possibly due to contamination, casting doubt on the reliability of these findings. It would be important for the authors to validate that the H92Q mutant is indeed GTPase-active but specifically deficient in Pi release.

Finally, several technical issues warrant attention. The quality of the smFRET and stopped-flow data for 30S head rotation does not appear sufficient to support strong conclusions. In Fig. 2C (middle panel), the donor and acceptor traces exhibit little anticorrelation, making it difficult to confidently interpret the FRET signal. Similarly, in Supp. Fig. 7, the trace is very noisy, and the final signal does not return to the start value, as would be expected for complete forward and reverse 30S head rotations. The EF-G binding trajectories in Supp. Fig. 8 also appear unusually noisy. Additionally, it remains unclear how EF-Tu behaves in experiments using non-hydrolysable GTP analogs—does it also exchange nucleotides, similarly to EF-G, and if so, how does this affect the experiments?

Overall, while the experiments are valuable and the results broadly align with existing models, the Introduction and Discussion need further clarification and focusing. Addressing the concerns outlined above—particularly the interpretation of GTP hydrolysis and Pi release, the sequence of conformational changes, and the technical quality of key data—would significantly strengthen the manuscript.

Reviewer #2

(Remarks to the Author)

The manuscript of Van Patten et al. is an important piece of biophysics that explores the mechanism by which the ribosome unwinds mRNA structure during translocation. This has been an area of debate, in particular the role of GTP hydrolysis in the translocation step and in the unwinding of RNA secondary structures during translocation. Bustamante and co-workers have previously performed elegant optical trapping and correlated fluorescence-force (fleezer) experiments to outline the general dynamics and energetics of translation elongation through a long hairpin. Here they expand these approaches using the fleezer assay correlated with hairpin unwinding using a variety of GTP concentrations and analog mixtures as well as mutant EF-Gs that change hydrolysis or phosphate release. The authors study the effect of GTP hydrolysis and phosphate release from EF-G on the unwinding of an mRNA hairpin during elongation using a combined optical tweezers/fluorescence microscopy technique (fleezers) by using GTP analogues and EF-G mutants. These first experiments beautifully correlate EF-G binding events with hairpin unwinding in 3 nt steps (shown previously). Their results show that GTP hydrolysis or phosphate release does not have any effect on hairpin unwinding but does affect the departure of EF-G from the ribosome and return of the ribosome to the post-translocation state. Their results suggest that tight EF-G binding leads to hairpin unwinding and subsequent mRNA movement based on previous structures and the fact that they see futile “loose” EF-G binding events that are not dependent on GTP hydrolysis. In a parallel set of fleezer experiments, they use doubly-labeled 30S subunits to correlate hairpin unfolding with head-body relative movement and here see a correlation with FRET changes (head rotation) and unwinding. Finally, they perform a distinct force measurement, pulling on the 30S subunit directly on an unstructured mRNA where translocation extends the tether, and correlate these events (under force) with EF-G binding under different conditions as above for the hairpin unwinding.

The results here clearly support their model whereby GTP hydrolysis is required only to release a tightly bound EF-G from the ribosome, where the translocation/unwinding etc is driven by the free energy of EF-G binding to the ribosome. This is an important work in the biophysics of translation and absolutely deserves publication in Nature Communications. For the most part it is clearly written and presented and the data are rigorously analyzed. I have some suggestions that should hopefully help the authors improve the presentation.

1. For the head to body FRET experiments, S12 to S19, the data are very noisy. This probably a highly dynamic conformation, as also shown by Fig sup 7 bulk FRET. I do not believe single-molecule FRET data has ever been presented, so I would have liked to see the behavior of this signal in TIRF under a range of conditions. Have the authors measured any FRET, which would allow the reader to better understand the Fleezer experiment.
2. Some of the figures are hard to follow, in particular the schematic in Figure 2. Its hard to see the double dye position. Maybe the authors could improve this figure, and also add a structure figure to supplemental data to show the dye positions in the non-rotated and rotated head states.
3. Fig 4 schematic free energy diagram is critical, and I wonder how the relative magnitudes for the free energies were calculated. There is no scale on the z-axis, but there is the free energy of peptide bond formation and GTP hydrolysis (where did these values come from? The PT value seems too small to me.
4. One key aspect of the model is the idea of EF-G “tight binding”. It would be nice if the authors could expound more on this state in the text and what is the free energy of this binding interaction. Do they have data that at least put bounds on the difference from the weak (sampling) and “tight” state in terms of lifetime/free energy?

Love this work--Jody Puglisi

Reviewer #3

(Remarks to the Author)

In this paper, Van Patten et al. use single-molecule approaches to investigate the role of GTP turnover by EF-G in driving ribosome translocation during translation. Using a hairpin unwinding assay with fluorescent EF-G, they show that slowing GTP hydrolysis/ Pi release elongate EF-G binding after, but not before, each codon unwinding event. Using an assisting-force translocation assay with fluorescent EF-G, they show that slowing GTP hydrolysis/ Pi release elongates EF-G binding both before and after a one-codon translocation. They conclude unwinding and translocation are distinct events, associated with a "tightening" isomerization event and head rotation, respectively; neither event relies on GTP hydrolysis or Pi release. They propose that the energy from GTP is instead used for undoing the "tightening" isomerization, as needed to proceed to the next cycle. The single-molecule Fleezers experiments are intricate, and the data quality are excellent. Overall, I find that the claims made are relatively well-justified by the data provided, and think that the impact will be well-appreciated in the translation and single-molecule fields. However, I do have some concerns, listed below, about some assay details and more generally about statistics from the very small data sets acquired. I recommend publication after addressing these issues in revision.

Major concerns

1. The arguments in this work depend heavily on the priors that GDPNP is nonhydrolyzable and that GTP γ S is slowly hydrolysable by EF-G. These are likely true assumptions, but it would strengthen the work to test them directly using the exact constructs tested in the single-molecule assays. A commercially available phosphate detection assay (e.g. Malachite green) would suffice.
2. Fig. 2 data are not particularly convincing. If noise is a limiting factor (line 224-225), and averaging was used to improve baselines, then why were only N=27 steps analyzed? I do not understand how the 20 ms estimate was made. Furthermore, there are essentially no claims made from the data (even the temporal correlation claim is later negated). Getting this experiment to work would strongly improve the impact of this paper. However, if revisiting this experiment to find conditions with cleaner FRET signal is out of the question, then an alternative option is to remove the figure from the manuscript entirely (lines 85-88 similarly).
3. Fig. 3: I am concerned as to whether or not translation is occurring here. Please provide more raw data in the main figure space. Fig. S8 only shows single steps -- was there any indication of processive translation in these experiments? If not, and they are indeed only single steps, what evidence is there that the observed steps are actually coupled with translation (was a peptide bond made)? If peptide synthesis is not occurring, I am not convinced that the results are meaningful. Peptidyl transfer is repeatedly identified as a key part of the overall thermodynamics in this system.
4. In both assay types used in this work (hairpin and assisting force), the applied force contributes to the kinetics measured. The authors should investigate whether the trends they observe are independent of applied force by testing more than one force level. Particularly since a goal of the work is to investigate underlying driving forces.

Minor concerns

1. Fig. 1b, why do ~half the steps not have an associated fluorescence peak? Is the ribosome stepping without EF-G? Are a subset of the EF-G molecules unlabeled?
2. Please provide details in main/methods for how errors on characteristic times were determined. Are they 95% CI from bootstrapping the CDF fit? Simple CI from fitted residuals may underestimate error in CDF fitting.
3. Towards the point above, the N event reported throughout this work are low. Typically 60-90 events, of which less than half contribute to the interpretable phase of the bi-exponential fits. Some are ~N=20. I appreciate the difficulty/labor of these complex experiments, and am reluctant to request higher N even though I think it would strengthen the work. Minimally, please explain assay limitations in the main or methods, and justify your acceptance of these very small sample sizes..
4. Fig. 1c, why is the sample mean reported, rather than 1/k 2 ? The fast rate constant is interpreted as the GTP rate for the mixed nucleotide experiments, and k 2 should be within error for all experiments with GTP as such. It is unclear why "WT" matches a bi-exponential, and what the slow phase may represent. Please explain.
5. Fig. 2 may be difficult to read for individuals who are red-green colorblind
6. Methods: Translation mix lists valine – specific components. Is this true? Synthesizing poly-valine? Is this important? Please provide more details.
7. Supp. Table 1: Only one of the three GFPNP conditions ran has its fit reported. Table should be the same size as Supp. Table 2.

Version 1:

Reviewer comments:

Reviewer #1

(Remarks to the Author)

In their revised manuscript, the authors have improved the technical quality of the data in Fig. 2d, increasing the number of events from n = 27 in the original manuscript to ~400, and they have introduced a few additional explanations. However, they did little to revise the text in response to my critical comments. I therefore reiterate my main concerns here, in slightly

different words, to make my points absolutely clear.

1. Novelty of the conclusions

The novelty is limited to the claim that mRNA unwinding is independent of GTP hydrolysis or Pi release. While this is possible, it is not sufficient to justify the energy diagram presented in Fig. 4. Moreover, it remains unclear whether the pulling force applied by optical tweezers destabilizes RNA structures to the point where unwinding becomes independent of external factors. This possibility is not adequately addressed. The other central messages—that translocation can occur without GTP hydrolysis and that the effect of GTP hydrolysis is kinetic—are not new; they have been reported repeatedly over the past 30 years. As such, the manuscript does not resolve any of the existing controversies.

2. Placement of Pi release in the energy diagram

The most critical issue is the order of steps in the authors' energy diagram. By placing Pi release after translocation, the authors implicitly assign Pi release as the trigger for EF-G dissociation. Their own data provide no direct evidence for this placement, and it contradicts the bulk of the literature. In particular, the cited cryo-EM structures (Petrychenko et al. and Carbone et al.) consistently show EF-G in the GDP-Pi state before tRNAs reach the chimeric hybrid state, and in the GDP state once the hybrid state is reached. This strongly argues against positioning Pi release after tRNA movement, as the authors do in Fig. 4. (The Rundlet et al. study [ref. 23] is not informative, as their assignment of a GDP-Pi intermediate as GTP is a misinterpretation of cryo-EM density.)

Additional kinetic studies from the Rodnina and Cooperman groups, which measured Pi release directly and compared it with defined steps of translocation, are in full agreement with the cryo-EM studies. These data are either ignored or downplayed by the authors, despite being directly relevant. Correctly placing Pi release at the step of tRNA movement (consistent with the measured kinetic effect) would yield a model fully consistent with existing cryo-EM data and with the prevailing loaded-spring GTPase model, in which Pi release triggers conformational rearrangements of EF-G that bias tRNA movement and accelerate translocation. In my opinion, the model in Fig. 4 must be revised accordingly: Pi release should be placed before tRNA movement to the chimeric states, and the corresponding energy barrier for translocation should be lowered.

3. EF-G H92Q mutant

The interpretation of the results obtained with the EF-G H92Q mutant hinges entirely on its biochemical activity. Does this mutant, unlike other H92 variants, hydrolyze GTP but fail in Pi release? This must be tested explicitly by assaying GTP hydrolysis and Pi release. Without such validation, the interpretation of the mutant data is not reliable.

4. Misrepresentation of the literature

The manuscript misrepresents earlier models for how GTP hydrolysis and Pi release drive translocation. While terms such as “power stroke” or “lever arm” have been used, these models (originating from multiple groups, not only Rodnina et al., as the citations suggest) were never meant to imply changes in the ground state. Rather, they consistently explain the kinetic effects of GTP hydrolysis in terms of lowering the transition barrier. This is clearly visible in the energy diagrams in the cited studies. By writing in the Introduction that “the energy of GTP hydrolysis and Pi release is directly converted to mechanical work,” the authors misrepresent the intent of prior work.

5. Abstract statement on thermodynamic requirement

Given the problems outlined above regarding the placement of Pi release, I question the validity of the Abstract's statement that “...while GTP hydrolysis kinetically accelerated translocation, it is thermodynamically required only to liberate the tightly bound EF-G from the ribosome.” The overall trajectory may differ from the one the authors propose, and in fact alternative models consistent with this possibility have already been suggested in the literature (e.g., ref. 36).

6. Description of the tRNA trajectory

A remaining issue concerns the Introduction's description of tRNA translocation. The authors state that swiveling of the 30S head domain moves the tRNA and mRNA into chimeric hybrid states, which is correct. However, they omit that this step is accompanied by a backward movement of the 30S body domain. This feature is well established from cryo-EM structures (refs. 18, 22) and stopped-flow data, and it is an essential element of the translocation mechanism. It should be explicitly included.

Reviewer #2

(Remarks to the Author)

The authors have nicely and rigorously addressed my concerns, and also those of the other reviewer. The manuscript is now acceptable for publication in Nature Comm.

Reviewer #3

(Remarks to the Author)

The authors have addressed all my questions and suggestions appropriately. This is a wonderful manuscript, and I recommend it for publication.

REVIEWER COMMENTS

Author responses are in blue.

Reviewer #1 (Remarks to the Author):

The paper by Van Patten et al. presents experiments that simultaneously monitor several steps of tRNA–mRNA translocation on the ribosome using high-resolution optical tweezers with fluorescence detection, a method validated in their previous work. This technique enables the reconstruction of the order of events, including mRNA unwinding relative to EF-G binding, mRNA unwinding versus 30S head rotation, and mRNA translocation versus EF-G binding.

We thank the reviewer for their detailed comments and helpful suggestions. We have addressed individual remarks below. Descriptions of updates made to the manuscript are indicated in **boldface** below. All changes in the manuscript DOCX file itself have been made with ‘Track Changes’ enabled.

While the experiments appear robust overall, aside from some technical issues noted below, the authors do not fully address the questions they initially pose. For example, they begin by asking about the GTP requirement for translocation but then discuss their data in terms of the ground-state thermodynamics of tRNA movement—specifically, whether the energy from GTP hydrolysis directly drives translocation by stabilizing the ground state of reaction intermediates. The distinction between the “mere acceleration” of translocation and a “requirement” for GTP hydrolysis is difficult to follow, as translocation can proceed spontaneously, even in the absence of GTP or EF-G. Thus, it is unclear why “requirement” is interpreted in terms of ground-state stabilization while “acceleration” is (correctly) associated with lowering the transition state barrier.

We feel there may have been a misunderstanding and we appreciate the opportunity to clarify the questions addressed in the paper. In Introduction, we asked the “fundamental question of whether the energy from GTP hydrolysis and Pi release are required to drive the energetically uphill events during ribosomal translocation.” (paragraph 3, last sentence.) This question directly aims at the energetic contribution of GTP hydrolysis/Pi release; specifically, at what time is this contribution required during translocation? The energetic requirement is best interpreted in terms of an energy landscape, where

free energy differences between intermediates determine whether transitions between them are thermodynamically favorable. For example, if GTP hydrolysis was energetically required for mRNA unwinding, transition to the unwound state would be unfavorable (could not occur irreversibly) unless in the presence of GTP hydrolysis. The energetic requirement is distinct from kinetic effects, where an already favorable transition can be accelerated (by lowering the transition barrier) or not, depending on the presence of GTP for example. Also, as the reviewer mentioned, and as we also stated in Introduction (paragraph 3, line 9), “translocation can proceed spontaneously, even in the absence of GTP or EF-G, albeit at very slow rates and under special conditions” but this has only been observed on unstructured mRNA. While this observation indicates that the free energy released as a result of peptidyl transfer (estimated 3-4kcal/mol by Liu et al, ref 7) is sufficient to favor unidirectional translocation on unstructured mRNA, the presence of stable hairpins changes the energy balance significantly. **We have updated the Introduction and Discussion text to clarify these points.** As the updated text indicates, hairpins can be as stable as ~9kcal/mol per codon, so the energy landscape in the presence of stable hairpins may be unfavorable overall unless the GTP hydrolysis contribution (~12kcal/mol) is also supplied. Importantly, however, this does not automatically mean that *prior* GTP hydrolysis is directly required for unwinding and translocation in each translocation cycle. Rather, in principle, another exergonic transition (such as EF-G tight binding) can first supply the energy to unwind the hairpin, with GTP hydrolysis being required only later to reverse the transition and reset the system. Indeed, our results support exactly this type of scenario, which we discuss in the manuscript.

The literature referenced in the manuscript consistently indicates that GTP hydrolysis and subsequent Pi release reduce the transition state barrier for mRNA translocation and promote EF-G release from the ribosome. Earlier kinetic studies suggested that Pi release, rather than GTP hydrolysis per se, is the key step, and recent cryo-EM structures (refs. 18 and 22) demonstrate how Pi release is coupled to tRNA translocation. Most models interpret the effects of Pi release in terms of a reduced transition state barrier, with little evidence supporting the idea that GTP hydrolysis stabilizes or destabilizes the ground state or “drives energetically uphill events” (c.f. p. 2). Power-stroke models, which the authors mention, focus on rate acceleration and, additionally, propose that conformational changes following Pi release prevent backward tRNA movement. There is no doubt that EF-G binding plays an important role in translocation, but the contribution of GTP hydrolysis and Pi release is essential, as they bring the rate of translocation to physiologically relevant levels. When these literature findings are compared with the data presented in Fig. 3—where replacing GTP with a non-hydrolysable analog increases both $\tau(\text{Pre})$ (the time to translocation) and $\tau(\text{Release and Post})$ (due to impaired EF-G release)—there is no apparent contradiction. Instead, the results are entirely consistent with established models. This section would benefit from revision to clarify this point and avoid suggesting inconsistencies where none exist.

There are inconsistencies with some previous interpretations in the sense that kinetic results had been interpreted by some investigators (e.g., in refs 4,15,17,18,35,39,40) to mean that the energy of GTP hydrolysis or Pi release is what drives the mechanical translocation of mRNA (also necessarily its

unwinding if base-paired), summarized as “Converting GTP hydrolysis into motion” (title of ref 17). We did not intend to dispute the important and well-known kinetic role of EF-G-mediated GTP hydrolysis in accelerating translocation. Rather, our results clarify and emphasize the separation between kinetic and energetic contributions of GTP hydrolysis to specific events during translocation. This is because, unlike most previous studies, we can directly probe mechanical work performed by individual ribosomes during translocation.

Translocation involves multiple intermediates with transition states between them. While GTP hydrolysis/Pi release accelerates the pathway as a whole, it is important to determine which intermediates and transitions are specifically affected. And the effect on each intermediate could be energetic (stability of the intermediate), kinetic (height of barrier to that intermediate), or both. In this manuscript, we particularly examined the effect on unwinding of mRNA and on mRNA-body translocation. We found that unwinding is not affected at all (kinetically or thermodynamically); furthermore, mRNA-body translocation still occurs without hydrolysis, meaning it is still thermodynamically favorable although kinetically slow. This is inconsistent with the view that mRNA translocation and unwinding are driven primarily by GTP hydrolysis/Pi release energy.

To avoid suggesting inconsistencies where they do not exist, **we have amended the text in the last section of Results, and also changed the title of that section to emphasize the well-known kinetics aspect. We have also added a note in Discussion (paragraph “Our findings enable ..”) to highlight the distinction between causal and temporal relationships.** We are grateful to the reviewer for the clarifying suggestion.

We will address the cryo-EM structure interpretations in response to another comment further below.

A second concern relates to the authors' claims regarding the role of GTP hydrolysis and Pi release in stabilizing the ground state of the post-translocation complex. The paper does not provide direct evidence for this statement, as it relies solely on rate measurements, making it difficult to distinguish between ground-state stabilization/destabilization and changes in the transition state barrier. As it is well established that without GTP hydrolysis, EF-G forms a stable complex with the ribosome, however this could reflect a high transition state barrier for EF-G dissociation rather than a true ground state stabilization. Additionally, it is well established that many tRNA pairs on the ribosome exhibit a favorable thermodynamic gradient for forward translocation. However, whether the tRNA(Val) used in this study shares this property remains unclear, and the authors may wish to experimentally assess whether the forward direction is indeed thermodynamically favorable.

Our depiction of the stabilization of the post-translocation state due to GTP hydrolysis and Pi release is based on thermodynamic principles rather than direct observation: when the GTP hydrolysis reaction ($\text{GTP} + \text{H}_2\text{O} \rightarrow \text{GDP} + \text{Pi}$) is coupled to translocation, the ΔG between the pre- and post-translocation states of the system with hydrolysis will differ from that without hydrolysis by precisely the amount of

the free energy change of the hydrolysis reaction. (It is always the free energy of the entire system of reactants and products that is considered.) **We have updated the Discussion paragraph (“The energy expenditure of ..”) which we hope better clarifies this point.**

We agree with the reviewer that based solely on our results, lengthening of EF-G release time could in principle be kinetic (not thermodynamic). However, the bulk studies showing stable ribosome-EF-G binding in the absence of GTP hydrolysis (e.g., in complex with GTP analogs) as mentioned by the reviewer, provide direct evidence for this claim, as they are performed under equilibrium or near-equilibrium where a small dissociation constant (stable complex) corresponds to thermodynamically unfavorable unbinding.

Different tRNAs have indeed been shown to have different propensities for forward translocation, and it has been suggested that the spontaneous factor-free translocation observed for poly-U mRNA (poly-Phe) may not be reproducible in other contexts. Although we wish to study the effect of different tRNAs, not just valine tRNA, in future studies, we note that the conclusions of this manuscript are unaffected by any tRNA-specific difference. In fact, a smaller degree of favorability or even unfavorability of translocation without EF-G (black path in Fig 4b) as alluded to by the reviewer, would strengthen our argument. We further expand on this point in response to the next comment.

There is also a potential issue with the energy profile presented in Fig. 4. The model begins with a pre-translocation complex (two tRNAs in the A and P sites) and ends with a complex containing only a tRNA in the P site—a state expected to be destabilized due to the loss of A-site interactions. This raises the question of why the final state (N+1) appears to have a lower energy than the pre-translocation state (N) after EF-G dissociation in the presence of GTP. A more detailed discussion of this aspect would improve clarity. Additionally, performing single-codon translocation experiments with tRNAs that thermodynamically favor backward translocation could provide further insight into the role of GTP hydrolysis.

We thank the reviewer for highlighting this issue. Indeed, the pre- and post-translocation states depicted in Fig 4 only span the translocation phase of the elongation cycle, and do not represent the same state in successive elongation cycles. Their ΔG is therefore not that of the entire elongation cycle which is on average equal to $\Delta G_{\text{peptidyltransfer}}$. **We have thus replaced $\Delta G_{\text{peptidyltransfer}}$ with $\Delta G_{\text{intrinsic}}$ in Fig 4b which allows it to have any value.** Because of the loss of A-site interactions as the reviewer mentioned, $\Delta G_{\text{intrinsic}}$ could be less negative than $\Delta G_{\text{peptidyltransfer}}$ (closer to zero or even positive for some tRNAs; see previous response). **We have also amended the corresponding text in the Discussion paragraph (“Accordingly, a conceivable ..”) to improve clarity on this point.**

We note that the pre-translocation complex follows peptidyl transfer and corresponds to the hybrid-state conformation. Thus, we do not start with two tRNAs in the A and P sites in Fig 4, but rather with one tRNA in the A/P state and the other in the P/E state. The P/E tRNA is very weakly bound, so there

would be no large loss of binding energy when it moves to the E site. Therefore, the main energy event is movement of the A/P tRNA to the P site (P/P state). Its CCA end does not change much on the 50S subunit, so the change in binding energy is essentially that of the anticodon stem loop (ASL). The ASL moves from the 30S A site to the 30S P site, which are similarly strong binding sites. In fact, it is believed that the P/P state is stronger than the A/P state. Together, these considerations would agree qualitatively with a negative total free energy change depicted in Fig. 4.

If anything, taking into account the tRNA binding energy differences and having a diminished (or even positive) overall ΔG between pre and post states in the uncatalyzed (black) path in Fig 4b would bolster, rather than weaken, the two major qualitative takeaways from our proposed landscape: (1) the proposal that EF-G tight binding drives mRNA unwinding and mRNA-body translocation will be bolstered, because the role of GTP hydrolysis is eliminated by our results and the other source of energy (now denoted $\Delta G_{\text{intrinsic}}$) can now be more easily excluded, and (2) the idea that EF-G-ribosome binding is thermodynamically trapped unless rescued by GTP hydrolysis is also bolstered, because now there is no source of energy other than GTP hydrolysis to help favor EF-G release.

Therefore, we believe that while performing translocation with other tRNAs will be informative, they are not essential for the arguments presented in the manuscript.

The section on mRNA helix unwinding is also somewhat unclear. The data suggest that unwinding occurs early and is independent of both GTP hydrolysis and Pi release, aligning with the authors' previous work in Molecular Cell. However, the overlap with their earlier findings and the novelty of these results are not clearly delineated, and the final mechanistic interpretation of unwinding remains somewhat speculative. Similarly, the observation that EF-G can bind and unbind from the ribosome before productive engagement has been reported in several smFRET studies and is already incorporated into most translocation models. Its inclusion in this study appears to serve completeness rather than introduce novel mechanistic insight.

The unwinding results are entirely novel and form a major contribution of this work. Our previous study published in Molecular Cell only employed WT conditions with GTP and did not address the role of GTP hydrolysis. The antibiotic fusidic acid was employed in that study to prevent EF-G release, which lengthened τ_{release} as expected. The present study is the first to directly examine the effect of GTP hydrolysis and Pi release on single-molecule unwinding kinetics, and we used various perturbations including the use of GTP analogs and EF-G and ribosome mutants in the unwinding assay. While the mechanism of unwinding remains speculative, the results are nevertheless highly significant as they show that the ribosome-EF-G complex can readily and rapidly unwind a stable hairpin (a significant amount of mechanical work) regardless of whether GTP can be hydrolyzed. **We re-worded the Results text (first section, second paragraph) to highlight that the effect of GTP hydrolysis on $\tau_{\text{unwinding}}$ and τ_{release} was previously unknown.**

It is true that unproductive EF-G binding has been reported, as we also cite in the manuscript. Here we have reported what we observed in our experimental traces. Furthermore, observation of unproductive binding and its insensitivity to GTPase perturbations are significant because they support our interpretation that for the ribosome to commit to translocation, EF-G must undergo a tight binding transition beyond the initial loose form of binding. Since tight binding can perform work (is exergonic), the insight from these observations is that hairpin unwinding (which requires work) can be driven by the free energy available from the EF-G tight binding.

In addition, the terminology used to describe conformational changes is confusing. The authors refer to “30S head rotation” without clarifying whether this term is equivalent to “swiveling” or how it relates to the formation of the chimeric state—a key intermediate in translocation, as shown by the Noller group.

As suggested, **we have added clarification in the Introduction (first paragraph, tracked changes)** to note that 30S head rotation refers to the same motion known as “swiveling” and that it produces the chimeric hybrid tRNA states.

Cryo-EM studies (refs. 18 and 22) indicate that tRNA movement into the chimeric state requires Pi release. Notably, there is no disagreement between these two papers; rather, both differ from ref. 23 (as discussed in refs. 18 and 22). Given these findings, the order of events in Fig. 4 (where complete forward head rotation is shown before Pi release) appears to be reversed. Since Pi release is not measured directly in this study, the rationale for this ordering remains unclear.

We believe that the structural data in refs 18, 22, and 23 are not conclusive whatsoever regarding whether 30S head rotation (chimeric hybrid state formation) *requires* Pi release. The set of ribosome configurations observed in these three studies are essentially the same. In all three, a head-non-rotated Pi-retaining state and a head-rotated Pi-released state are observed. Since two correlated changes (head rotation and loss of Pi) are seen at the same time, it is not possible to establish temporal precedence and therefore it is not possible in these studies to establish a causal connection. Indeed, Petrychenko et al (ref 18) interpreted the results to mean Pi release triggers the rotation, Carbone et al (ref 23) interpreted the opposite order, while Rundlet et al (ref 22) took a more agnostic position. With our powerful single-molecule assays, we have direct evidence that abolishing GTP hydrolysis and Pi release does not abolish completion of mRNA-body translocation (corresponding to head rotation). This result, which is consistent with recent stopped-flow experiments by Rexroad et al (ref 24), is presented and elaborated on in the final section of Results and in the Discussion. It indicates that at least under GTPase-perturbed conditions, head rotation happens before/without GTP hydrolysis and Pi release.

Whether the same order also holds under normal conditions is not known; we have therefore remained cautious in Fig 4 and indicated “Pi release (if delayed)” (rather than just “Pi release”) on the dashed line separating states III and IV. Also, the red gradient bar at the bottom of Fig 4a marks the likely timeline of “GTP hydrolysis and Pi release (normally fast if unperturbed).” As the reviewer mentioned, we do not monitor Pi directly, and we do not claim to time this event independently from GTP hydrolysis using our results, except that it can occur only after hydrolysis. To further emphasize this, **we now mention in Discussion (paragraph “Our findings enable ..”) to indicate that Pi release may temporally precede head rotation when not delayed.** We thank the reviewer for bringing this issue to our attention.

The discussion also relies on findings with the EF-G H92Q mutant, which is based on prior work of a different lab suggesting that this mutant retains GTPase activity but has reduced Pi release (ref. 33). However, concerns arise from the fact that the supposedly GTPase-deficient H92A mutant still shows significant GTPase activity (ref 33), possibly due to contamination, casting doubt on the reliability of these findings. It would be important for the authors to validate that the H92Q mutant is indeed GTPase-active but specifically deficient in Pi release.

We can only speculate on why the estimated GTPase activity for H92A was higher in Koripella et al (ref 33) compared to earlier studies (e.g., Cunha et al, ref 39) but the possibility of WT contamination does not necessarily negate the findings regarding the other mutant H92Q which clearly appears defective. As the reviewer noted earlier, we have no direct measure of GTP hydrolysis or Pi release in our assay. In our interpretations, we have accordingly treated the two chemical events together, one simply following the other. Therefore, our conclusions do not rely on the validity of measurements by Koripella et al (ref 33). This is particularly the case since we observe qualitatively similar results for H92Q and H92A in our assays. Had we seen drastically different results, then the contrast between H92A and H92Q would have been more relevant.

Finally, several technical issues warrant attention. The quality of the smFRET and stopped-flow data for 30S head rotation does not appear sufficient to support strong conclusions. In Fig. 2C (middle panel), the donor and acceptor traces exhibit little anticorrelation, making it difficult to confidently interpret the FRET signal. Similarly, in Supp. Fig. 7, the trace is very noisy, and the final signal does not return to the start value, as would be expected for complete forward and reverse 30S head rotations. The EF-G binding trajectories in Supp. Fig. 8 also appear unusually noisy. Additionally, it remains unclear how EF-Tu behaves in experiments using non-hydrolysable GTP analogs—does it also exchange nucleotides, similarly to EF-G, and if so, how does this affect the experiments?

Our smFRET assay is a tour de force experiment that involves co-temporally monitoring two single-molecule channels, the smFRET channel and the optical tweezers channel, during active translation for the first time. The low signal to noise is in part due to the small FRET change (only ~ 0.3) and the short-lived low-FRET state. Despite noisy traces, we see clear anticorrelation between donor and acceptor emissions, as shown in Fig 2c where not only the green donor signal increases near unwinding steps but the red acceptor signal also decreases correspondingly. The correlation coefficient for the section shown is -0.7 . Importantly, the conclusion we draw from the smFRET assay is merely that there is a temporal correlation between unwinding and forward head rotation (FRET decrease) under normal conditions. We believe the smFRET results strongly support such a correlation. To strengthen this claim, **we have now added a large number of new data for the event averaging shown in Fig 2d, from $n=27$ in the original manuscript to $n\sim 400$** , which shows FRET decrease near the unwinding steps in a much more convincing manner. **We have also updated Supp Fig 7 with additional panels.** We believe the new analysis strengthens our work and we thank the reviewer for the constructive criticism.

While the raw data from the stopped-flow measurements shown in Supp Fig 7 (now panel b) are naturally noisy, they fully agree with previously published results (refs 24 and 37) and show a clear trend (an initial rapid decrease followed by a slow recovery) with kinetic parameters that match those measured previously. This result is included for completeness in our study to show that our Atto dyes produce the same behavior as the Alexa dyes used in the previous works. The interpretation of data and the kinetic parameters (rate and amplitude of forward and reverse head rotation curves) had been done in the cited references and we did not feel obliged to repeat those for this control experiment, as the original authors of those publications are also co-authors in this paper.

The assisting force data (in Supp Fig 8) are noisier compared to the unwinding assay because of smaller step size (3 vs 6 nt) and larger dynamics due to ribosome tethering. However, we can still see clear steps and EF-G binding events. **In the new Fig 3 (panel g), we have added a trajectory with multiple steps** to demonstrate that processive translocation by exact 3nt (codon) steps is taking place. **We have also added a multi-step panel to Supp Fig 8 for the GDPNP condition, clearly showing (long) EF-G binding events.**

As mentioned in Methods, EF-Tu was pre-incubated with GTP and loaded with tRNA to form a ternary complex before being added to the translation mix containing the GTP analogs. We expect the long half-life of ternary complex to limit the effect of the analogs on EF-Tu, but we cannot rule out that this factor may have also been affected. If EF-Tu activity is affected by the analogs, the time between consecutive unwinding steps (outside of the EF-G-Cy3 binding dwell time) would be expected to increase, for which we have seen no evidence. Importantly, since we focus on the EF-G dwell time in our analysis of translocation, such an increase would not affect our measurements or our conclusions regarding translocation. **We have now noted the possibility that EF-Tu may be affected in Results (paragraph "GDPNP is ..").**

Overall, while the experiments are valuable and the results broadly align with existing models, the Introduction and Discussion need further clarification and focusing. Addressing the concerns outlined above—particularly the interpretation of GTP hydrolysis and Pi release, the sequence of conformational changes, and the technical quality of key data—would significantly strengthen the manuscript.

We thank the reviewer again for all the feedback. We have made several changes in the Introduction and Discussion sections of the manuscript that we hope have helped with clarity and focus.

Reviewer #2 (Remarks to the Author):

The manuscript of Van Patten et al. is an important piece of biophysics that explores the mechanism by which the ribosome unwinds mRNA structure during translocation. This has been an area of debate, in particular the role of GTP hydrolysis in the translocation step and in the unwinding of RNA secondary structures during translocation. Bustamante and co-workers have previously performed elegant optical trapping and correlated fluorescence-force (fleezer) experiments to outline the general dynamics and energetics of translation elongation through a long hairpin. Here they expand these approaches using the fleezer assay correlated with hairpin unwinding using a variety of GTP concentrations and analog mixtures as well as mutant EF-Gs that change hydrolysis or phosphate release. The authors study the effect of GTP hydrolysis and phosphate release from EF-G on the unwinding of an mRNA hairpin during elongation using a combined optical tweezers/fluorescence microscopy technique (fleezers) by using GTP analogues and EF-G mutants. These first experiments beautifully correlate EF-G binding events with hairpin unwinding in 3 nt steps (shown previously). Their results show that GTP hydrolysis or phosphate release does not have any effect on hairpin unwinding but does affect the departure of EF-G from the ribosome and return of the ribosome to the post-translocation state. Their results suggest that tight EF-G binding leads to hairpin unwinding and subsequent mRNA movement based on previous structures and the fact that they see futile “loose” EF-G binding events that are not dependent on GTP hydrolysis. In a parallel set of fleezer experiments, they use doubly-labeled 30S subunits to correlate hairpin unfolding with head-body relative movement and here see a correlation with FRET changes (head rotation) and unwinding. Finally, they perform a distinct force measurement, pulling on the 30S subunit directly on an unstructured mRNA where translocation extends the tether, and correlate these events (under force) with EF-G binding under different conditions as above for the hairpin unwinding. The results here clearly support their model whereby GTP hydrolysis is required only to release a tightly bound EF-G from the ribosome, where the translocation/unwinding etc is driven by the free energy of EF-G binding to the ribosome. This is an important work in the biophysics of translation and absolutely deserves publication in Nature Communications. For the most part it is clearly written and presented and the data are rigorously analyzed. I have some suggestions that should hopefully help the authors improve the presentation.

We are grateful to the reviewer for the helpful and constructive criticism. We have addressed individual remarks below. Descriptions of updates made to the manuscript are indicated in **boldface** below. All changes in the manuscript DOCX file itself have been made with ‘Track Changes’ enabled.

1. For the head to body FRET experiments, S12 to S19, the data are very noisy. This probably a highly dynamic conformation, as also shown by Fig sup 7 bulk FRET. I do not believe single-molecule FRET data has ever been presented, so I would have liked to see the behavior of this signal in TIRF under a range of conditions. Have the authors measured any FRET, which would allow the reader to better understand the Fleezer experiment.

As the reviewer mentioned, the noisy FRET signal is likely caused in part by highly dynamic conformations. The signal-to-noise ratio is very low and the predicted FRET change is only ~ 0.25 (from 0.25 to 0.5). We have made some attempts to measure FRET using TIRF microscopy under different conditions, but observed similar noisy traces with the additional disadvantage of not knowing if the FRET signals came from functional ribosomes and when any translocation steps actually occur. On Fleezers, we can focus on the translocation steps and identify FRET change patterns (by event averaging, as shown in Fig 2d) which helps eliminate dynamics that occur outside of the translocation timeframe.

2. Some of the figures are hard to follow, in particular the schematic in Figure 2. Its hard to see the double dye position. Maybe the authors could improve this figure, and also add a structure figure to supplemental data to show the dye positions in the non-rotated and rotated head states.

As suggested, **we have updated the schematic in Fig 2a to better show the dye positions. We have also amended Supp Fig 7 with a panel showing labeling positions as suggested.**

3. Fig 4 schematic free energy diagram is critical, and I wonder how the relative magnitudes for the free energies were calculated. There is no scale on the z-axis, but there is the free energy of peptide bond formation and GTP hydrolysis (where did these values come from? The PT value seems too small to me.

Please note that in the updated Fig 4b we have replaced $\Delta G_{\text{peptidyl transfer}}$ with $\Delta G_{\text{intrinsic}}$, because $\Delta G_{\text{peptidyl transfer}}$ drives the entire elongation cycle, and ΔG for the translocation phase alone may differ from that. That said, free energy available from peptidyl transfer ($\sim 3-4$ kcal/mol estimated in Liu et al – ref 7) and GTP hydrolysis ($\sim 10-12$ kcal/mol estimated based on physiological concentrations) are approximately represented. As suggested by the reviewer (in this and the next comment), **we have made a more quantitative description of the landscape in the main text including the above numbers.** However, to avoid giving the impression that Fig 4b is an exact quantitative depiction of the energy landscape, we find it best not to add the values to the vertical axis. The diagram is intended to only depict general and qualitative view of the three pathways and their comparison.

4. One key aspect of the model is the idea of EF-G “tight binding”. It would be nice if the authors could expound more on this state in the text and what is the free energy of this binding interaction. Do they

have data that at least put bounds on the difference from the weak (sampling) and “tight” state in terms of lifetime/free energy?

This is a very interesting question. Our data and the resulting model proposed in Fig 4 indeed allow some estimation of the lower and upper bounds for the tight binding energy. On the one hand, the tight binding transition cannot be so strong as to not be rescuable by GTP hydrolysis (~10-12 kcal/mol), and on the other hand it cannot be so weak as to not afford (along with $\Delta G_{\text{intrinsic (I-II)}}$ indicated in Fig 4b) the unwinding of stable hairpins (up to ~9 kcal/mol/codon based on nearest-neighbor energies). These set the upper bound at ~10-12 kcal/mol and the lower bound at slightly below 9 kcal/mol. **We have now included a brief presentation of these ideas in the text.**

Love this work--Jody Puglisi

Thank you again Jody! We appreciate your helpful comments.

Reviewer #3 (Remarks to the Author):

In this paper, Van Patten et al. use single-molecule approaches to investigate the role of GTP turnover by EF-G in driving ribosome translocation during translation. Using a hairpin unwinding assay with fluorescent EF-G, they show that slowing GTP hydrolysis/ Pi release elongate EF-G binding after, but not before, each codon unwinding event. Using an assisting-force translocation assay with fluorescent EF-G, they show that slowing GTP hydrolysis/ Pi release elongates EF-G binding both before and after a one-codon translocation. They conclude unwinding and translocation are distinct events, associated with a “tightening” isomerization event and head rotation, respectively; neither event relies on GTP hydrolysis or Pi release. They propose that the energy from GTP is instead used for undoing the “tightening” isomerization, as needed to proceed to the next cycle. The single-molecule Fleezers experiments are intricate, and the data quality are excellent. Overall, I find that the claims made are relatively well-justified by the data provided, and think that the impact will be well-appreciated in the translation and single-molecule fields. However, I do have some concerns, listed below, about some assay details and more generally about statistics from the very small data sets acquired. I recommend publication after addressing these issues in revision.

We wish to thank the reviewer for the insightful comments. We have addressed individual remarks below. Descriptions of updates made to the manuscript are indicated in **boldface** below. All changes in the manuscript DOCX file itself have been made with ‘Track Changes’ enabled.

Major concerns

1. The arguments in this work depend heavily on the priors that GDPNP is nonhydrolyzable and that GTP γ S is slowly hydrolysable by EF-G. These are likely true assumptions, but it would strengthen the work to test them directly using the exact constructs tested in the single-molecule assays. A commercially available phosphate detection assay (e.g. Malachite green) would suffice.

The behavior of these analogs has been well studied. Strong evidence for GDPNP being nonhydrolyzable comes from numerous previous studies including those on ribosomes (e.g., Rodnina et al, ref 4). Observation of GDPNP at high occupancy in the EF-G GTPase active site in crystal structures of the ribosome-EF-G complexes that are obtained days to weeks after complex formation (Zhou et al, ref 50) is another type of strong evidence.

We note that although we used these assumptions in our arguments to motivate the GTP:GDPNP titration experiments (Fig 1f) or to simplify the reasoning (Supp Fig 10), the major conclusions in the manuscript hold even without these assumptions. For example, even if GDPNP is hydrolyzed at some slow rate, our titration results still provide evidence for nucleotide exchange, which means the rate of GDPNP hydrolysis (if it occurs at all) must be slower than the timescales relevant to our experiments. To emphasize this, **we have made changes the relevant text in the second section of Results.**

Regarding the reviewer's suggested malachite green assay, the main issue is that we include both GTP and the analog in our translocation mixes (described in second section of Results) to allow multi-turnover translation with both EF-Tu and EF-G present. A simple phosphate measurement assay under such multi-turnover conditions with a mix of GTP+analog will yield highly convoluted results, where GTP hydrolysis would strongly dominate the Pi generation. We cannot use only GDPNP without GTP, as that will limit translocation to a single turnover (ref 4). What can be done is single- or multi-turnover GTP hydrolysis assay using vacant ribosomes (which is the standard in the field), and as mentioned above, this has been reliably done previously by other research groups using rapid flow kinetics.

As a control, we used the malachite green assay suggested by the reviewer to compare the extent of Pi generation by GTP, GTPγS, and GDPNP under our buffer conditions. Malachite green has a color development time of ~30 minutes, making it suitable mainly for end-point assays. After 30 minutes, malachite green absorption at 620nm was observed above the background for all samples to different extents likely due to residual Pi present. Afterward, overnight incubation at 37C resulted in a marked increase in absorption for GTP and to a lesser extent for GTPγS, but no increase was detected for GDPNP. We therefore see no reason to doubt the validity of previous results regarding these analogs.

2. Fig. 2 data are not particularly convincing. If noise is a limiting factor (line 224-225), and averaging was used to improve baselines, then why were only N=27 steps analyzed? I do not understand how the 20 ms estimate was made. Furthermore, there are essentially no claims made from the data (even the temporal correlation claim is later negated). Getting this experiment to work would strongly improve the impact of this paper. However, if revisiting this experiment to find conditions with cleaner FRET signal is out of the question, then an alternative option is to remove the figure from the manuscript entirely (lines 85-88 similarly).

In response to the reviewer's comment, we are happy to report that **we have collected a much larger set of data for the event averaging shown in Fig 2d, from n=27 in the original manuscript to n=396, resulting in a more convincing FRET signature. We have also amended the text to include analysis of FRET change kinetics allowed by the larger data set, and added new panels to Supp. Fig. 7 accordingly.** We believe that these updates have strengthened our work and we thank the reviewer for the constructive criticism.

It should be noted that, as indicated in the manuscript, the temporal correlation seen in Fig 2 is not necessarily indicative of a *causal* relationship; and it is the causal relationship that is later negated under GTPase perturbations; the temporal correlation under normal conditions remains valid. **The last paragraph of this section is updated to emphasize this distinction.**

While no strong claim is made from the FRET results regarding causality, the results themselves are highly significant because they show temporal correlation between forward head rotation and mRNA hairpin unwinding under normal conditions. Prior to performing this experiment, it was suspected but

not confirmed whether forward, as opposed to reverse, head rotation correlates with unwinding, for which these results provide a resolution at the single-molecule level for the first time.

3. Fig. 3: I am concerned as to whether or not translation is occurring here. Please provide more raw data in the main figure space. Fig. S8 only shows single steps -- was there any indication of processive translation in these experiments? If not, and they are indeed only single steps, what evidence is there that the observed steps are actually coupled with translation (was a peptide bond made)? If peptide synthesis is not occurring, I am not convinced that the results are meaningful. Peptidyl transfer is repeatedly identified as a key part of the overall thermodynamics in this system.

In response to the reviewer's comment, **we have added a new panel g in Fig. 3 depicting a multi-turnover translocation trace to show that we are observing genuine translation (and peptide bond formation by implication). We have also added a similar trace for the GDPNP condition in the Supp Fig. 8g.**

The issue that limits the throughput of the assisting-force assay is not the lack of translocation. It is rather that most steps are so noisy and buried in back-and-forth ribosome dynamics that their exact timing and therefore the τ_{pre} and τ_{post} measurements become too uncertain. We commonly see stepping and fluorescent events, but due to the noise, we are unable to confidently analyze them all. So, the reviewer's valid concern is fully accounted for, and it was also our concern when implementing this assay for the first time. Since we see multi-turnover stepping, with exact codon (3nt) step sizes (as seen in the new Fig 3g, and more clear when data is lowpass-filtered), and concomitant with EF-G binding, we feel confident that this is natural translocation. Furthermore, the translocation response to perturbations in the assisting force assay for the most part mirrors that in the unwinding assay which provides another control for validity of this assay.

4. In both assay types used in this work (hairpin and assisting force), the applied force contributes to the kinetics measured. The authors should investigate whether the trends they observe are independent of applied force by testing more than one force level. Particularly since a goal of the work is to investigate underlying driving forces.

We agree that the applied force can affect the observables. Indeed, our previous studies have already shown that the overall translocation rate in the unwinding assay (Qu et al – ref 45), and the time between EF-G binding and hairpin unwinding in this assay ($\tau_{\text{unwinding}}$; Desai et al – ref 32) vary with force, as does the overall translocation rate in an opposing force assay (Liu et al – ref 7). The case is simpler for hairpin unwinding where the force directly affects only the mRNA. For the assisting force assay, the

relationship is more complicated. Collecting data at different forces, especially for the GTPase-perturbed conditions would be extremely challenging given the throughput. This is because lowering the assisting force from the current range (9-12pN) would exacerbate the extension noise. And, at higher forces we observe more frequent slipping and less stepping.

To overcome these limitations and address the force dependence, we intend to replicate the current results in the opposing force geometry in the future. By flipping the direction of the force relative to translocation, we will be able to make more definite statements regarding the energetics of the process. However, we believe that the current conclusions remain valid regardless of the extent of force sensitivity.

Minor concerns

1. Fig. 1b, why do ~half the steps not have an associated fluorescence peak? Is the ribosome stepping without EF-G? Are a subset of the EF-G molecules unlabeled?

Steps without an observed binding signal could indeed be unlabeled EF-G (labeling efficiency is in the 90% range) or labeled-but-bleached EF-G. In general, we detect a discernible fluorescence signal for 70-80% of steps in this assay.

2. Please provide details in main/methods for how errors on characteristic times were determined. Are they 95% CI from bootstrapping the CDF fit? Simple CI from fitted residuals may underestimate error in CDF fitting.

We initially used nonlinear least-squares fittings of the raw data and did not use bootstrapping; the errors were listed as 95% confidence intervals from fits, as was indicated in figure and table legends.

In light of the reviewer's comment, we investigated other ways to estimate uncertainty, and we think a more rigorous and reliable alternative to the ordinary least squares method and the nonparametric bootstrapping error estimation is fitting using maximum likelihood estimation (MLE) which has a well-defined likelihood and error model for single and bi-exponential data. When we fit our titration data using MLE, uncertainties were higher than those from least squares fitting, but nevertheless validated our conclusions, as shown in the new Supp Tables. **We have now updated the main text, the methods section, Fig 1, Supp Fig 3, and Supp Tables 1-3 to reflect the fitting of data using MLE.**

We recognize that more data could always improve our results, but we still strongly believe we have sufficient data from those particular titration experiments to make a case.

3. Towards the point above, the N event reported throughout this work are low. Typically 60-90 events, of which less than half contribute to the interpretable phase of the bi-exponential fits. Some are $\sim N=20$. I appreciate the difficulty/labor of these complex experiments, and am reluctant to request higher N even though I think it would strengthen the work. Minimally, please explain assay limitations in the main or methods, and justify your acceptance of these very small sample sizes..

We acknowledge the reviewer's concern and agree that larger sample sizes would provide more robust estimates. Sample sizes were limited by the low throughput of the experiments especially those under GTPase perturbed conditions, requiring up to months of data collection per condition. Nevertheless, given the magnitude of the effects observed and the type of claims made, we believe that the sample sizes were adequate. In Supp Table 4, we show the results of statistical tests using methods that are valid for small samples such as those obtained here. **We have also amended the Methods (under "Fitting of distributions") with a description of limitations.**

4. Fig. 1c, why is the sample mean reported, rather than $1/k_2$? The fast rate constant is interpreted as the GTP rate for the mixed nucleotide experiments, and k_2 should be within error for all experiments with GTP as such. It is unclear why "WT" matches a bi-exponential, and what the slow phase may represent. Please explain.

Starting with the last question, the bi-exponential fit of τ_{release} for WT traces (also noted in Desai et al, 2019, ref 32) indicates that after mRNA unwinding there must be two major kinetic paths for EF-G release. It is indeed unclear what these paths represent, and if and how the GTP analogs affect them together or individually. We cannot therefore assume that the k_2 (fast) rate for analogs necessarily matches k_2 for GTP. **We have added a note in Results (first section, first paragraph) clarifying that the cause for the bi-exponential distribution of τ_{release} is unknown.**

In Fig 1c, sample mean was reported for the WT condition to capture the overall rate. **For consistency and clarity, we have updated Fig 1c to show fast and slow rates for the WT+GTP case as well. We have also updated Fig 1 legend to clarify the error bars in each case.**

5. Fig. 2 may be difficult to read for individuals who are red-green colorblind

Thank you for noticing this accessibility issue. **We have made sure that the essential features in the new updated Fig 2 are visible to the red-green color blind (see below).**

6. Methods: Translation mix lists valine – specific components. Is this true? Synthesizing poly-valine? Is this important? Please provide more details.

Yes, after synthesizing a 9-aa leader sequence containing Met, Asp, Tyr, and Lys residues, the rest of the translated region (~50 aa which is translated inside the Fleezers chamber) is made entirely of valine residues. This or very similar mRNA designs have been used in previous single-molecule works (refs 2, 7, 32, and 45) and the exact sequence in the present work is from Desai et al (ref 32). While poly-Val is a non-natural sequence, these designs are well-behaved in bulk and single-molecule translation assays. For experimental simplicity and also for streamlining data analysis and interpretations, we have kept the same model mRNA design in this study. **We have added a clarification regarding mRNA under Method heading “Preparation of mRNA and tRNA”.**

7. Supp. Table 1: Only one of the three GFPNP conditions ran has its fit reported. Table should be the same size as Supp. Table 2.

Thank you for noticing this. **We have included these conditions in the updated Supp Table 1 as suggested.**

REVIEWER COMMENTS

Reviewer #1 (Remarks to the Author):

We thank all the reviewers for the constructive criticism. Author responses to the remaining points are written in blue. Descriptions of changes to the manuscript are in **boldface**. All new changes in the manuscript DOCX file have been tracked.

In their revised manuscript, the authors have improved the technical quality of the data in Fig. 2d, increasing the number of events from $n = 27$ in the original manuscript to ~ 400 , and they have introduced a few additional explanations. However, they did little to revise the text in response to my critical comments. I therefore reiterate my main concerns here, in slightly different words, to make my points absolutely clear.

1. Novelty of the conclusions

The novelty is limited to the claim that mRNA unwinding is independent of GTP hydrolysis or Pi release. While this is possible, it is not sufficient to justify the energy diagram presented in Fig. 4.

We respectfully disagree with the assessment that the novelty of this work is limited to the results of the unwinding assay. While this section of our results was perhaps the most unexpected, the other two major sets of experiments (the smFRET head rotation assay and the assisting-force mRNA-body translocation assay) yielded important results that corroborated our proposal regarding the energy requirement and its origin during translocation. First, the smFRET results directly and clearly measured for the first time the temporal correlation between 30S head rotation (mRNA-body movement) and mRNA unwinding during processive translocation under conditions in which GTP hydrolysis is not perturbed. Second, the assisting force assay results enabled us to distinguish mRNA-body movement from the mRNA unwinding step as separate events that, although correlated under normal GTP hydrolysis conditions, can become uncoupled under GTPase perturbation conditions. Third, the combination of these results yielded the conclusions that the unwinding step is independent of GTP hydrolysis both energetically and kinetically, and that mRNA-body translocation depends on GTP hydrolysis only kinetically but not thermodynamically. Together, these results constitute the main contribution of our work, results that have been obtained through the design of unprecedented single-molecule experiments, and that allowed us to dissect individual endergonic steps in the

translocation pathway and to assign their relative free energies and barriers. The diagram presented in Fig 4 reflects this analysis and is intended to illustrate qualitatively the type of energy landscape that is consistent with the experimental findings. **We have now edited all three panels of the Fig 4 and their captions for clarity; we have explicitly mentioned in the caption to panel b: “A qualitative model, consistent with experimental results, of the energy landscape of translocation under normal conditions (in red) and compared qualitatively with non-canonical conditions (without GTP hydrolysis or without EF-G altogether, in green and black, respectively).”**

Moreover, it remains unclear whether the pulling force applied by optical tweezers destabilizes RNA structures to the point where unwinding becomes independent of external factors. This possibility is not adequately addressed.

We had already addressed this point explicitly with references in the penultimate paragraph in Discussion: “Since the optical tweezers pulling force alone is insufficient to unwind the codon duplex without the ribosome’s assistance [refs 2, 32, 48], an exergonic change of state in the ribosome complex upon EF-G binding must be involved [..]” Our previous work has shown that the force applied by the optical tweezers certainly does not destabilize the hairpin to the point where unwinding becomes independent of external factors. The need for mechanical work performed by the ribosome to unwind mRNA is the foundational basis of this assay, which was established over a decade ago and thoroughly investigated under a range of forces (see refs 2, 32, 48). **We have now amended the Methods text (under “optical tweezers measurements”) to clarify this point: “These forces were chosen to be sufficiently high to provide single-codon resolution, and in the case of the unwinding assay, also far below the critical force [refs 2,32,48] of the hairpin so that hairpin unwinding would not occur spontaneously without the participation of the active ribosome (Supp. Fig. 1c).”**

We have also included a new panel in Supp Fig 1 (and also copied below) showing an example trajectory in which a hairpin held under an applied force of ~14 pN in front of an inactive ribosome remains folded for more than 10 minutes after which we terminate the experiment. As a figure directly derived from experimental data is worth more than the proverbial thousand words, if our results were due to an artefact of the force applied to the hairpin rather than to the activity of the ribosome, we would still be gathering the data.

c

No activity

The other central messages—that translocation can occur without GTP hydrolysis and that the effect of GTP hydrolysis is kinetic—are not new; they have been reported repeatedly over the past 30 years. As such, the manuscript does not resolve any of the existing controversies.

Our work is the first to demonstrate: (1) the temporal coincidence of hairpin unwinding and head rotation (by the smFRET assay), (2) the breakdown of this correlation under GTPase perturbation (by the assisting-force translocation assay), (3) that the specific steps of mRNA unwinding and mRNA-body movement within a translocation cycle *do not require* GTP hydrolysis, and (4) that GTP hydrolysis can, nonetheless, have a kinetic (catalytic) effect on the mRNA-body movement. These results are completely novel and represent a significant advance over what had been established previously in the field, namely that, as a whole, “translocation can occur without GTP hydrolysis and that the effect of GTP hydrolysis is kinetic.” **We have now amended the text in the third paragraph of Discussion, explaining our argument regarding the relevance of GTP hydrolysis energy (or lack thereof) in mRNA/tRNA movement: “Importantly, even if an event such as mRNA-body movement is accelerated as a result of GTPase activity, it is not necessarily the case that the energy of GTP hydrolysis is used in that acceleration. Rather, EF-G (after hydrolysis) can act as a catalyst, and acceleration will result simply from its preferential binding to, and stabilization of the transition state immediately preceding the event [new ref 41]. Clearly, the energy of this binding is unrelated to any energy released up to this event by hydrolysis. Thus, the kinetic acceleration observed in the presence of GTP hydrolysis does not imply that it utilized the energy resulting from hydrolysis.”**

2. Placement of Pi release in the energy diagram

The most critical issue is the order of steps in the authors' energy diagram. By placing Pi release after translocation, the authors implicitly assign Pi release as the trigger for EF-G dissociation. Their own data provide no direct evidence for this placement, and it contradicts the bulk of the literature. In particular, the cited cryo-EM structures (Petrychenko et al. and Carbone et al.) consistently show EF-G in the GDP-Pi state before tRNAs reach the chimeric hybrid state, and in the GDP state once the hybrid state is reached. This strongly argues against positioning Pi release after tRNA movement, as the authors do in Fig. 4. (The Rundlet et al. study [ref. 23] is not informative, as their assignment of a GDP-Pi intermediate as GTP is a misinterpretation of cryo-EM density.)

As we label in Fig 4b, and as our previous response to the reviewer's original remarks indicated (remarks already reflected in the revised text in the second paragraph of Discussion), we placed the Pi release strictly after mRNA-body translocation *only* when GTP hydrolysis is perturbed by EF-G mutants or GTP analogs (labeled "if delayed" in Fig 4). On the other hand, in the same figure, we clearly label the occurrence of Pi release under normal conditions as possibly taking place any time from as early in the pathway as before mRNA-body translocation, to a variable time after the latter. This would be in line with the branched kinetic pathway put forth originally by the Rodnina group (ref 15), where it was proposed that Pi release precedes tRNA movement in one branch and follows it in the other. **In the new version of the paper, we now make this point explicit in Fig 4a and in its caption: "Note that under normal conditions, hydrolysis and Pi release can occur before or after mRNA-body translocation (bottom, red gradient), but they are strictly required only to progress to state IV which is followed by EF-G dissociation (red arrow)."**

Furthermore, we are in disagreement with the reviewer on the conclusiveness of cryoEM results regarding the timing or causality of Pi release relative to mRNA-body translocation (i.e., head rotation and formation of chimeric hybrid states). The cryoEM snapshots in which two changes occur together (Pi release and head rotation) do not establish temporal or causal order at all. **We have amended the text in the second paragraph of Introduction to highlight this caveat: "Specifically, P_i release is seen concomitant with conformational changes that move the mRNA relative to the 30S body domain. Strictly speaking, it is not possible to establish a causal relationship between these two correlated events from structural snapshots. Indeed, the three studies did not agree on when GTP hydrolysis and P_i release are required during translocation."**

We have directly examined causation by perturbing Pi release, and the scheme in Fig 4 simply reflects our results. This type of causality relations has been made possible by the use, for the first time, of correlated single-molecule force and fluorescence spectroscopies.

Additional kinetic studies from the Rodnina and Cooperman groups, which measured Pi release directly and compared it with defined steps of translocation, are in full agreement with the cryo-EM studies. These data are either ignored or downplayed by the authors, despite being directly relevant. Correctly placing Pi release at the step of tRNA movement (consistent with the measured kinetic effect) would yield a model fully consistent with existing cryo-EM data and with the prevailing loaded-spring GTPase model, in which Pi release triggers conformational rearrangements of EF-G that bias tRNA movement and accelerate translocation. In my opinion, the model in Fig. 4 must be revised accordingly: Pi release should be placed before tRNA movement to the chimeric states, and the corresponding energy barrier for translocation should be lowered.

As stated above, in our model, GTP hydrolysis and Pi release *can* occur before mRNA-body movement (or equivalently, chimeric hybrid state formation). The energy barrier to this movement *is* lowered by GTP hydrolysis, as already indicated in Fig 4b (see the difference between green and red landscapes at the boundary between states II and III). **Following the reviewer's suggestion, we have now magnified this difference in Fig 4b to further highlight the kinetic effect.** However, our results clearly demonstrate that Pi release *can* also occur after mRNA-body movement.

3. EF-G H92Q mutant

The interpretation of the results obtained with the EF-G H92Q mutant hinges entirely on its biochemical activity. Does this mutant, unlike other H92 variants, hydrolyze GTP but fail in Pi release? This must be tested explicitly by assaying GTP hydrolysis and Pi release. Without such validation, the interpretation of the mutant data is not reliable.

We addressed the reviewer's original comment regarding EF-G H92Q mutant previously. Our interpretation does not hinge at all on whether H92Q can hydrolyze GTP (and is only defective in Pi release) or if it is mainly defective in GTP hydrolysis. This is partly because our results for

H92Q and the hydrolysis-deficient H92A mutant are qualitatively similar, thus precluding us from making any specific statement about Pi release. Please see Supp Fig 10, where we explain that H92Q defect in GTP hydrolysis would be also consistent with our proposed model (in which hydrolysis and Pi release are not strictly required for mRNA-body translocation).

Furthermore, the reviewer previously questioned the validity of the findings of Koripella et al (ref 33) on H92Q based on apparent contradiction between Koripella's characterization of another mutant, H92A, and the prior characterization by Rodnina's group of that same mutant (ref 39). The two groups reported different values for the absolute GTPase activity for the H92A mutation ($\sim 28 \text{ s}^{-1}$ and undetected, respectively) which can be due to different experimental details and does not mean that the qualitative results obtained for this and other mutants studied by Koripella are wholly invalid. The reviewer had previously suggested that Koripella's H92A results may be due to contamination with wild-type EF-G, implying that such a situation could invalidate any similar characterization by the Koripella's group. This is an experimentally unsupported statement, and we point out that while it is possible for apparent hydrolysis to result from wild-type EF-G contamination, it is not easy to rationalize how WT contamination could result in reduced Pi release in H92Q. Incidentally, Koripella's work was published in *Scientific Reports* in 2015 and has not since been challenged publicly.

4. Misrepresentation of the literature

The manuscript misrepresents earlier models for how GTP hydrolysis and Pi release drive translocation. While terms such as "power stroke" or "lever arm" have been used, these models (originating from multiple groups, not only Rodnina et al., as the citations suggest) were never meant to imply changes in the ground state. Rather, they consistently explain the kinetic effects of GTP hydrolysis in terms of lowering the transition barrier. This is clearly visible in the energy diagrams in the cited studies. By writing in the Introduction that "the energy of GTP hydrolysis and Pi release is directly converted to mechanical work," the authors misrepresent the intent of prior work.

We wish to re-emphasize a distinction that may be helpful here. To say that the effect of GTP hydrolysis is an overall kinetic effect over-simplifies the nature of the process. Each translocation cycle is a multi-step process, not a single step. The kinetic effect of hydrolysis will be different depending on what part of the cycle is being considered. What may appear to be a kinetic effect on the pathway as a whole, could result from changes both in some transition states (maxima in the reaction pathway) as well as some ground-state intermediates (minima) and not others.

What we have shown is a kinetic effect of GTPase activity pinpointed precisely to the mRNA-body movement (transition from state II to state III in Fig 4), whereas the other endergonic event, hairpin unwinding (transition from I to II), is not affected at all. The ability to distinguish which events are affected kinetically by GTP hydrolysis has become possible through the use of our correlated single-molecule force and fluorescence spectroscopy experiments, and reflect the novelty of our results.

In the Introduction and Discussion, we specifically referred to works in which the authors have explicitly attributed the source of energy for tRNA movement to GTP hydrolysis:

For example, Rodnina et al (ref 4) state: “The function of EF-G is to translate chemical energy, derived from GTP hydrolysis, into directional molecular movement on the ribosome, that is, to function as a motor protein.”

Savelsbergh et al (ref 15) state: “These results indicate that the energy of GTP hydrolysis is utilized to promote the ribosome rearrangement [..]”

Rodnina et al (ref 17) state: “[..] EF-G uses the energy of GTP hydrolysis for a power stroke-like motion acting on the ribosome and the tRNAs.”

Petrychenko et al (ref 18) state: “Thus, the present PRE–EF-G–GDP–Pi and the CHI–EF-G–GDP structures represent key intermediates at the onset of translocation that show how the chemical energy from GTP hydrolysis and Pi release translates into the forward movement of the mRNA–tRNA complex.”

Chen et al (ref 35) state: “Our results suggest that EF-G serves as a force-generating motor via a power stroke in the early steps of translocation, supporting the proposal that energy released from GTP hydrolysis triggers EF-G conformational changes to promote translocation.”

Cunha et al (ref 39) state: “Thus, part of the energy of GTP hydrolysis is used to drive a conformational change of the ribosome that controls translocation [..]”

Holtkamp et al (ref 40) state: “the energy of EF-G–GTP binding alone is not sufficient to promote rapid movement on the 30S subunit. Rather, the [step] which entails both 30S translocation and tRNA movement on the 50S subunit from the PRE A/P2 state into the early POST–G state, is driven by GTP hydrolysis, which couples conformational rearrangements of EF-G to the engagement of domain 4 with the 30S codon-anticodon complex.”

It is worth noting that a power stroke mechanism would necessitate a lowering of both the transition state energy as well as the ground (final) state energy due to the power stroke force applied along the reaction coordinate. Therefore, in the literature when GTP hydrolysis is attributed as the energy source for a power stroke model, it naturally follows that GTP hydrolysis causes a lowered ground state. **In the third paragraph of Discussion, we have**

therefore now explicitly clarified that EF-G.GTP binding can catalyze translocation without the need for contribution of GTP hydrolysis energy: “Previous observations of rapid GTP hydrolysis by EF-G combined with results of kinetic fluorescence measurements with EF-G mutants and GTP analogs have led to a GTP hydrolysis-driven power-stroke model for translocation [refs 4,15,17,18,35,39,40]. This model posits that EF-G conformational changes, fueled by the energy of GTP hydrolysis, drive the mechanical movement of mRNA and tRNAs relative to the 30S body domain. However, the fact that hydrolysis normally occurs before an event during translocation does not necessarily imply that it is energetically required for the event. GTP hydrolysis presumably precedes both hairpin unwinding and mRNA-body translocation, but our results indicate that it is not required thermodynamically to drive either of these endergonic events. Importantly, even if an event such as mRNA-body movement is accelerated as a result of GTPase activity, it is not necessarily the case that the *energy* of GTP hydrolysis is used in that acceleration. Rather, EF-G (after hydrolysis) can act as a catalyst, and acceleration will result simply from its preferential binding to, and stabilization of the transition state immediately preceding the event [new ref 41]. Clearly, the energy of this binding is unrelated to any energy released up to this event by hydrolysis. Thus, the kinetic acceleration observed in the presence of GTP hydrolysis does not imply that it utilized the energy resulting from hydrolysis.”

We have also amended the text in the second paragraph of Introduction to avoid the impression of mutual exclusivity: “Two roles for GTP hydrolysis and P_i release by EF-G during translocation have been described. In the first role, the energy from GTP hydrolysis and P_i release is suggested to be directly converted to mechanical work [refs 4,15,17,18]. In the second role, this energy is instead used for reducing the affinity of EF-G to the ribosome, allowing for EF-G release and the resetting of the translational cycle [refs 9,19-25].”

5. Abstract statement on thermodynamic requirement

Given the problems outlined above regarding the placement of P_i release, I question the validity of the Abstract’s statement that “...while GTP hydrolysis kinetically accelerated translocation, it is thermodynamically required only to liberate the tightly bound EF-G from the ribosome.” The overall trajectory may differ from the one the authors propose, and in fact alternative models consistent with this possibility have already been suggested in the literature (e.g., ref. 36).

We are aware of alternative models proposed in the literature (e.g. ref 36). However, as stated above, our work pertains to specific individual steps in the translocation pathway, namely the endergonic steps of mRNA unwinding and complete accurate mRNA-body movement.

6. Description of the tRNA trajectory

A remaining issue concerns the Introduction's description of tRNA translocation. The authors state that swiveling of the 30S head domain moves the tRNA and mRNA into chimeric hybrid states, which is correct. However, they omit that this step is accompanied by a backward movement of the 30S body domain. This feature is well established from cryo-EM structures (refs. 18, 22) and stopped-flow data, and it is an essential element of the translocation mechanism. It should be explicitly included.

First paragraph of Introduction has been changed to include this feature: "Next, forward rotation or "swivel" of the 30S head domain, accompanied by a partial reversion of inter-subunit rotation, moves the mRNA and tRNAs relative to the body domain to produce chimeric hybrid tRNA states [refs 12,13]. This step is followed by the reverse 30S head rotation without carrying back the mRNA and tRNAs, which together with full reverse inter-subunit rotation, completes the mRNA/tRNA movement and resets the ribosome to the classical post-translocation state."

Reviewer #2 (Remarks to the Author):

The authors have nicely and rigorously addressed my concerns, and also those of the other reviewer. The manuscript is now acceptable for publication in Nature Comm.

Reviewer #3 (Remarks to the Author):

The authors have addressed all my questions and suggestions appropriately. This is a wonderful manuscript, and I recommend it for publication.